# Poleward migration of western North Pacific tropical cyclones related to changes in cyclone seasonality

Xiangbo Feng [1✉], Nicholas P. Klingaman[1] & Kevin I. Hodges[1]

The average location of observed western North Pacific (WNP) tropical cyclones (TCs) has shifted north over the last several decades, but the cause remains not fully understood. Here we show that, for the annual average, the observed northward migration of WNP TCs is related to changes in TC seasonality, not to a northward migration in all seasons. Normally, peak-season (July–September) TCs form and travel further north than late-season (October–December) TCs. In recent decades, related to less frequent late-season TCs, seasonally higher-latitude TCs contribute relatively more to the annual-average location and seasonally lower-latitude TCs contribute less. We show that the change in TC seasonality is related to the different responses of late-season and peak-season TC occurrence to a stronger Pacific Walker Circulation. Our findings provide a perspective on long-term trends in TC activity, by decomposing the annual-average statistics into seasonal components, which could respond differently to anthropogenic forcing.

[1] National Centre for Atmospheric Science and Department of Meteorology, University of Reading, Reading, UK. ✉email: xiangbo.feng@reading.ac.uk

The location at which tropical cyclones (TCs) reach their lifetime-maximum intensity (LMI) has migrated poleward in most ocean basins in recent decades[1,2]. The Western North Pacific basin (WNP; 0–60°N, 100–180°E), accounting for >30% of global TCs, shows the largest poleward migration rate for LMI of all ocean basins. In the WNP, the annual mean TC formation (cyclogenesis) location has also shifted poleward[3]. These migration trends are primarily based on satellite-era TC Best Track observations. Studies have attempted to attribute TC poleward migration to anthropogenic warming, both in historical general circulation model (GCM) climate simulations and in future projections based on climate-change scenarios[2,4,5]. One particular hypothesised mechanism by which anthropogenic climate forcing may influence TC tracks is tropical expansion, related to a meridional extension of the Hadley circulation and a poleward shift of climatic conditions favouring TC cyclogenesis and intensification[1,3,6–8]. However, for WNP TCs, climate models with anthropogenic forcing do not fully reproduce the observed poleward migration rate in the satellite era, with either no poleward migration or a much slower migration rate than observed[2,4,5,9]. This suggests that GCMs might miss or underrepresent some processes that are crucial to the observed long-term trend in average TC location, provided that anthropogenic forcing is the cause.

The disparity between GCMs and observations also highlights concerns about detecting a robust long-term trend in the relatively short observational record, including the confounding effects of interdecadal climate variability and variations in operational techniques for TC identification[8,10]. In the satellite era, although lifetime-relative intensity (e.g., LMI) estimates are relatively reliable, uncertainty remains in identifying TC occurrence and location, particularly for storms obscured by clouds or formed from monsoon depressions[11]. Moreover, uncertainty in identifying weak stages of the TC lifecycle (e.g., cyclogenesis) and weak TCs may be larger[8,12,13]. There are time-dependent inter-agency discrepancies for WNP TC estimates in the Best Track data produced by different meteorological agencies[14–16]. All these uncertainties could affect the detection and magnitude of migration trends of WNP TCs. Using multiple sources of Best Track data from different agencies can reduce the uncertainty caused by temporal and inter-agency variations in techniques. Further, here we use TC track data derived from a climate reanalysis, in which TCs are consistently identified throughout the record with an objective approach independent of Best Track, to increase our confidence in the trends initially detected in Best Track.

There are two possible pathways for a significant poleward shift in the annual-mean location of TC activity, which are not mutually exclusive: (1) the seasonal-mean latitude of TC activity in most seasons has shifted poleward, but TC seasonality (the seasonal TC frequency relative to the annual frequency) has remained constant; (2) TC seasonality has changed over time, but the seasonal-mean latitude has remained constant. The latter would produce an apparent shift of TC annual-mean latitude, even though the seasonal latitude has not changed, by altering the relative contributions of seasons with climatologically higher- or lower-latitude TCs to the annual-mean latitude. The relative importance of these two pathways may depend on whether the poleward shift of TC-relevant environmental factors is seasonally uniform, which would favour the first pathway, or seasonally varying, which would favour the second pathway. The poleward shift of climatic conditions[1,6,8,17] might be related to the observed large-scale tropical expansion[18,19], while the changes in TC seasonality could be driven by regional-scale changes in environmental conditions. The latter may arise from regional variations of conditions in response to climate change, beyond a uniform tropical expansion. Without comprehensively evaluating changes

in TC seasonality, simply attributing the meridional shift of TC annual location to anthropogenic forcing or tropical expansion can be misleading, as this may exaggerate the effect of the meridional shift of environmental conditions.

Here, we show that over the satellite era WNP TCs have systematically migrated north on average, shown by TCs both forming and moving further north, in the latest Best Track data from three national meteorological agencies. The observed significant migration of TCs is confirmed in an independent TC track dataset derived from a climate reanalysis. More importantly, we show that this northward migration can largely be attributed to a significant change in TC seasonality, rather than to a seasonally uniform migration. This indicates that the poleward migration of WNP TC tracks solely due to a seasonally uniform poleward shift of environmental conditions may be small, and may not be statistically identifiable during the modern observational period. We further show that the change in seasonality is associated with a stronger Pacific Walker Circulation, which has seasonally varying effects on WNP TC occurrence.

## Results

**Northward shift of the annual-mean position of TCs over the satellite era.** In this work, we first calculate linear trends of the annual-mean of TC latitudes using a variety of metrics: (1) all six-hourly points along tracks (all-track-points: all six-hourly track points in the TC lifetime), tropical cyclogenesis (genesis: the first track point at which maximum sustained winds reach 34 knots), LMI (the first track point at which TC reaches its LMI) and cyclolysis (lysis: the last track point in the TC lifetime); (2) the net transit distances of TC track (lysis minus genesis) and all-track-points (all-track-points minus genesis), and the net transit distances of developing (LMI minus genesis) and dissipating (lysis minus LMI) lifecycle phases. To consider the effect of recurvature of storms on poleward migration, we also classify TCs into recurving and straight-moving TCs and estimate the linear trends for each group. Detailed definitions of these TC metrics are in the "Methods" section. Because analysing all-track-points does not depend on TC intensity, once intensity exceeds the TC identification threshold, and because there is a large sample size (>500 track points for ~20 TCs per year, on average), the detected location trends for all-track-points are statistically more reliable than those for other metrics. We analyse two types of TC tracks for 1979–2018: Best Track and climate reanalysis (see the "Methods" section). Best Track data from three meteorological agencies (JTWC, Joint Typhoon Warning Centre; JMA, Japan Meteorological Agency; and CMA, China Meteorological Administration) create a multi-source ensemble of Best Track ("Best Track ensemble" hereafter). As TC uncertainty is large for weak storms[12], our Best Track analysis only includes TCs reaching the intensity of severe tropical storm or higher (maximum sustained winds ≥48 knots or ≥24 m/s). TC tracks objectively tracked from the ERA-Interim climate reanalysis during the same period are used as a dataset independent of Best Track. We further extend the Best Track ensemble back to the pre-satellite era (1951–1978), to highlight the strong effect of TC seasonality change on the trend detection, even in this more uncertain period for TC estimates. To eliminate the effects of climate variability on trend and variability detection, in our analysis, the impacts of the El Niño Southern Oscillation (ENSO) and Pacific Decadal Oscillation (PDO) are removed using multiple linear regression[2] (see the "Methods" section). Note that the statistical significance for either trend or correlation values are at the 95% confidence interval, unless stated otherwise (see the "Methods" section).

Table 1 lists the meridional trends of the WNP TC metrics over the satellite era (1979–2018) using the Best Track ensemble mean.

**Table 1 Linear trends in annual-mean latitude of tropical cyclones and seasonality components, in Best Track ensemble mean.**

| 1979–2018 | All-track-points | Genesis | LMI | Lysis | All-track-point transit distance (all-track-point-genesis) | Transit distance (lysis-genesis) | Developing phase transit distance (LMI-genesis) | Dissipating phase transit distance (lysis- LMI) |
|---|---|---|---|---|---|---|---|---|
| Total | 78 ± 31* | 40 ± 28* | 61 ± 30* | 100 ± 59* | 29 ± 25* | 60 ± 55* | 21 ± 21* | 39 ± 49 |
| $\delta_P$ | 43 ± 19* | 23 ± 16* | 28 ± 17* | 49 ± 27* | 16 ± 7* | 26 ± 16* | 5 ± 3* | 21 ± 14* |
| $\delta_{Lat'}$ | 26 ± 30** | 6 ± 27 | 20 ± 27 | 31 ± 54 | 13 ± 24 | 25 ± 49 | 14 ± 20 | 12 ± 44 |
| $\delta_{P'Lat'}$ | 9 ± 19 | 11 ± 15 | 13 ± 16 | 20 ± 26 | −1 ± 12 | 9 ± 24 | 2 ± 7 | 7 ± 22 |
| **1951–1978** | **All-track-points** | **Genesis** | **LMI** | **Lysis** | **All-track-point transit distance (all-track-point-genesis)** | **Transit distance (lysis-genesis)** | **Developing phase transit distance (LMI-genesis)** | **Dissipating phase transit distance (lysis-LMI)** |
| Total | 2 ± 87 | 76 ± 65* | 50 ± 81 | −52 ± 146 | −97 ± 73* | −128 ± 134** | −44 ± 48** | −118 ± 106* |
| $\delta_P$ | 34 ± 35** | 20 ± 23** | 21 ± 23** | 34 ± 39** | 8 ± 15 | 14 ± 25 | −1 ± 4 | 15 ± 23 |
| $\delta_{Lat'}$ | −20 ± 68 | 63 ± 55* | 38 ± 69 | −62 ± 133 | −88 ± 69* | −124 ± 132** | −41 ± 48 | −119 ± 103* |
| $\delta_{P'Lat'}$ | −12 ± 30 | −6 ± 19 | −9 ± 21 | −24 ± 51 | −17 ± 26 | −17 ± 42 | −2 ± 15 | −15 ± 36 |

Linear trends (km/decade) in annual-mean latitude and its three seasonality contributing terms ($\delta_P$, $\delta_{Lat'}$, and $\delta_{P'Lat'}$) for western North Pacific TC track metrics, in the Best Track ensemble mean, over two epochs (satellite and pre-satellite eras) of the whole period 1951–2018. The ensemble consists of Best Track data from the Joint Typhoon Warning Centre (JTWC), Japan Meteorological Agency (JMA) and China Meteorological Administration (CMA); as JMA Best Track data usually lack intensity records over the pre-satellite era, over this period the ensemble for lifetime-maximum intensity (LMI) and other related metrics (transit distances of developing and dissipating phases) only consists of JTWC and CMA data. The El Niño Southern Oscillation and Pacific Decadal Oscillation effects have been removed with a multivariate regression before estimating the trend. ± represents the 95% confidence interval of trend value. * indicates significant trend at the 95% confidence. ** indicates significant trend at the 90% confidence.

Figure 1a shows the timeseries of annual-mean latitude for all-track-points, and Supplementary Figs. 1a and 2a for genesis and LMI, respectively. In the Best Track ensemble mean, we find that not only have genesis (40±28 km/decade; ± representing the 95% confidence interval of trend value, with the statistical details in the "Methods" section) and LMI (61±30 km/decade) significantly migrated north over the last four decades, but so too have all-track-points (78±31 km/decade). Compared to previous studies[1–3], which used a shorter data record (until 2013), the migration rate of LMI here is similar, but the migration rate of genesis is smaller possibly due to the different definitions of genesis (genesis has been defined with a threshold wind speed of 40 knots in ref. [3], contrasting with the 34 knots used here). We also find a statistically significant northward trend in the annual-mean latitude of lysis (100±59 km/decade). Furthermore, there are significant positive trends in the net latitudinal transit distances of TC track (60±55 km/decade) and all-track-points (29±25 km/decade), suggesting the meridional expansion of TC tracks (increased northward travelling distance). The increase in net latitudinal transit distance is related to a longer travelling distance both in the developing phase (21±21 km/decade) and dissipating phase (39±49 km/decade, not significant at 95% confidence). We find that the signs of migration trends in these TC metrics are consistent across Best Track sources (Supplementary Table 1), but the trend values are uncertain. Inter-agency uncertainty is small in genesis and LMI trends, but the uncertainty is large in the trends of lysis and transit distances, for which JTWC data have the smallest or least significant values. Thus, in the satellite era, the discrepancies in measuring TC tracks among different meteorological agencies[14–16] can affect the magnitudes and robustness of migration trends in WNP TCs, especially in the dissipating phase.

We find similar northward trends for WNP TCs in ERA-Interim (Table 2, Fig. 1b and Supplementary Figs. 1b and 2b), mostly with larger rates than in the Best Track ensemble mean. These differences are related to our objective tracking method, which extends the storm lifetime to the pre-TC and post-TC stages (this can be seen in Fig. 2a and Supplementary Fig. 3c), causing a wider meridional distribution (further poleward and equatorward) of tracks with stronger interannual variability in the annual-mean latitude. The annual-mean latitudes in the Best Track ensemble mean and ERA-Interim tracks are significantly correlated, suggesting ERA-Interim can reproduce the interannual variability in TC location. From these datasets, we conclude that WNP TC tracks have systematically shifted northward over the last four decades, owing both to the northward shift of genesis and to TCs travelling further north.

**Changes in the TC seasonality**. Figure 2a–c shows the 3-month rolling climatology of frequency, relative frequency and latitude, for all-track-points of WNP TCs in the Best Track ensemble mean and ERA-Interim tracks. Similar seasonal cycles are seen in genesis and LMI (Supplementary Figs. 3a–c and 4a–c). The TC frequency and location exhibit distinct seasonal variations. The peak-season (July–September; JAS) TCs, which account for >50% of the annual TC frequency, form and travel further north than the late-season (October–December; OND) TCs. Over the last four decades, the monthly frequencies of TCs and all-track-points have significantly decreased for the late season (at 95% confidence interval), but there are no significant changes for other seasons (Fig. 2d and Supplementary Fig. 3d). We find that the trends are more robust in the relative frequencies of TCs and all-track-points (Fig. 2e and Supplementary Fig. 3e), both with the largest positive rate of 1.0 ± 0.8 %/decade for peak-season TCs and the largest negative rate of −1.0 ± 0.7 %/decade for late-season TCs.

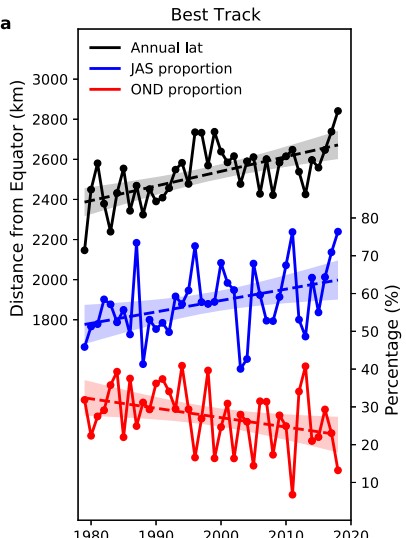
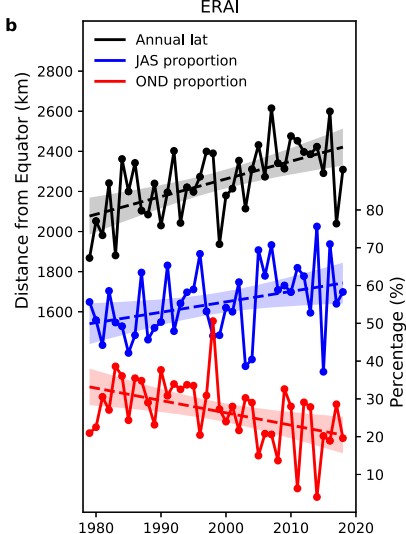

**Fig. 1 Changes in annual-mean latitude and seasonal relative frequency of tropical cyclone all-track-points. a** Timeseries of annual-mean latitude of all-track-points (black), and timeseries of peak-season (July–September; JAS, blue) and late-season (October–December; OND, red) relative frequencies of all-track-points in the year, over 1979–2018, in Best Track ensemble mean. **b** as a, but in the ERA-Interim TC dataset. Dashed lines represent the linear trend, with shading showing 95% confidence interval for the linear fit; the El Niño Southern Oscillation and Pacific Decadal Oscillation effects are removed by a multivariate regression; latitude is converted to distance from the equator (km).

**Table 2 Linear trends in annual-mean latitude of tropical cyclones and seasonality components, in ERA-Interim TC dataset.**

| 1979–2018 | All-track-points | Genesis | LMI | Lysis | All-track-point transit distance (all-track-points–genesis) | Transit distance (lysis–genesis) | Developing phase transit distance (LMI–genesis) | Dissipating phase transit distance (lysis–LMI) |
|---|---|---|---|---|---|---|---|---|
| Total | 88 ± 41* | 49 ± 30* | 59 ± 33* | 125 ± 98* | 41 ± 47** | 76 ± 108 | 11 ± 48 | 65 ± 77** |
| $\delta_{P'}$ | 36 ± 23* | 14 ± 14* | 20 ± 17* | 52 ± 36* | 24 ± 16* | 38 ± 28* | 6 ± 6** | 32 ± 24* |
| $\delta_{Lat'}$ | 23 ± 39 | 25 ± 27** | 26 ± 38 | 24 ± 112 | −2 ± 44 | −1 ± 119 | 1 ± 55 | −2 ± 83 |
| $\delta_{P'Lat'}$ | 30 ± 24* | 9 ± 17 | 13 ± 22 | 49 ± 56** | 19 ± 21** | 39 ± 57 | 4 ± 26 | 35 ± 46 |

Linear trends (km/decade) in annual-mean latitude and its three seasonality contributing terms ($\delta_{P'}$, $\delta_{Lat'}$ and $\delta_{P'Lat'}$) for western North Pacific TC track metrics, in the ERA-Interim TC dataset, over the satellite era (1979–2018). The El Niño Southern Oscillation and Pacific Decadal Oscillation effects have been removed with a multivariate regression before estimating the trend; ± represents the 95% confidence interval of trend value. * indicates significant trend at the 95% confidence. ** indicates significant trend at the 90% confidence.

For monthly latitudes, all-track-points have a northward trend for most seasons, but these trends are only statistically significant for September–November (Fig. 2f); neither LMI nor genesis has a significant northward trend in any season (Supplementary Figs. 3f and 4f). These seasonal changes are consistent between the Best Track ensemble mean and ERA-Interim TCs, despite a larger uncertainty in ERA-Interim possibly related to the wider meridional distribution of tracks and a smaller sample size.

The key to the seasonal redistributions of relative frequencies of TC and all-track-points is the decreasing frequency of late-season TCs (Supplementary Fig. 3d–e). We found no significant changes in TC duration for individual seasons or for the annual average, which demonstrates that seasonal changes in the number of all-track-points are related to changes in TC frequency, not to TC lifetime. In the Best Track ensemble mean, over the last four decades, the late-season TCs have decreased by −0.3 ± 0.2 TCs/decade in the monthly frequency. This dominates a decreasing trend of −1.5 ± 0.9 TCs/decade in the annual frequency. The peak-season TCs have a slightly decreasing but insignificant trend in frequency. However, when the peak-season frequency is divided by the annual frequency, which is decreasing, the resulting peak-season relative frequency has a statistically significant increasing trend. This suggests that the increasing trend in the peak-season relative frequency is driven by fewer TCs in the late season, and hence fewer TCs annually. In ERA-Interim, for peak-season TCs, the statistical significance of

changes in the relative frequency is also much higher than that in the frequency.

For illustration, Fig. 1 (blue and red lines) shows the 3-month relative frequency of all-track-points for peak-season and late-season TCs during 1979–2018. The peak-season relative frequency of all-track-points has increased steadily in both datasets, by >10% of the annual number over the four decades, which for the annual-mean location gives greater weight to peak-season TCs that are climatologically further north. In contrast, the late-season relative frequency of all-track-points has decreased by >10% of the annual number over the four decades, giving less weight to late-season TCs in the annual-mean location. As expected, a northward trend is seen in the annual-mean latitude (Fig. 1, black lines). For all-track-points, the annual-mean latitude is significantly correlated with both peak-season relative frequency ($r = 0.52$ and $0.31$ in the Best Track ensemble mean and ERA-Interim, respectively; both significant at 95% confidence) and late-season relative frequency ($r = -0.48$ and $-0.41$ in the Best Track ensemble mean and ERA-Interim, respectively). The discrepancies in the correlations between the two datasets suggest that in ERA-Interim the interannual variability of TC annual latitude is less determined by variations in TC seasonality. This is because ERA-Interim tracks include the post-TC stage of storms (after extratropical transition)[20], which travel along the mid-latitude westerlies with a small meridional displacement. Significant relationships are also found between seasonal relative

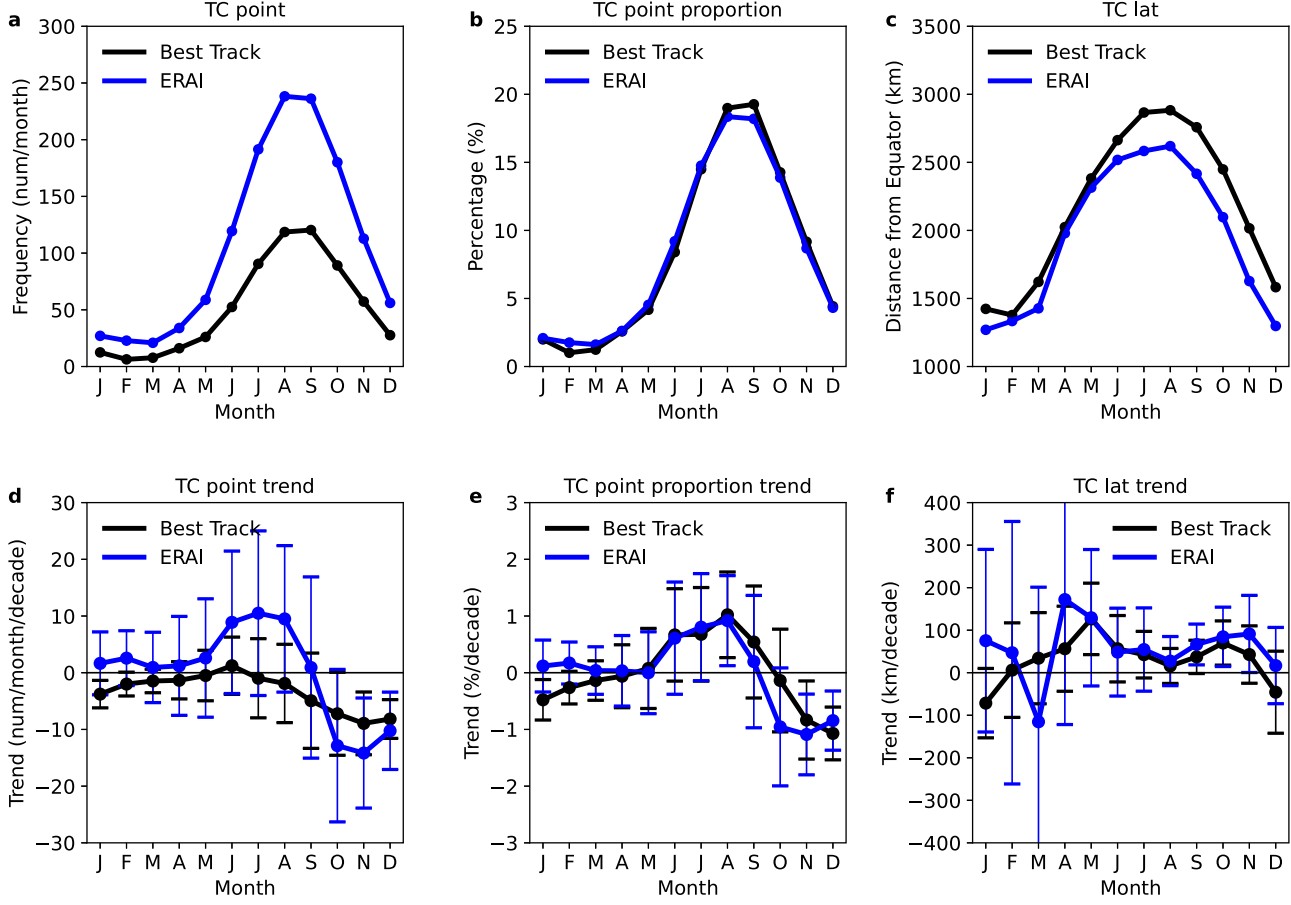

**Fig. 2 Seasonality of tropical cyclone all-track-points, and changes in seasonality. a–c** Climatology of 3-month rolling averages of monthly frequency, monthly relative frequency and monthly latitude for all-track-points, over 1979–2018, in the Best Track ensemble mean (black) and ERA-Interim (blue) TC datasets. **d–f** Linear trends in 3-month rolling averages of monthly frequency, monthly relative frequency, and monthly latitude for all-track-points, over 1979–2018, in the Best Track ensemble mean (black) and ERA-Interim (blue) TC datasets. In **c**, **f**, latitude is converted to distance from the equator (km); in the lower panels, the error bars show 95% confidence interval for the trends; the El Niño Southern Oscillation and Pacific Decadal Oscillation effects are removed by a multivariate regression.

frequency and annual-mean latitude for genesis and LMI (Supplementary Figs. 1 and 2).

**Decomposing the annual-mean latitude of TCs to seasonal components.** We decompose the annual-mean latitude of each TC metric into three components, to quantify the contributions to the long-term trends from the changes in monthly relative frequency ($\delta_{P'}$), monthly latitude ($\delta_{Lat'}$) and the covariance of these two monthly changes ($\delta_{P'Lat'}$). For details, see the "Methods" section. In short, for each year, $\delta_{P'}$ holds monthly latitude constant (at climatology) but retains the trend and interannual variability in monthly relative frequency; $\delta_{Lat'}$ holds monthly relative frequency constant (at climatology) but retains the trend and interannual variability in monthly latitude; $\delta_{P'Lat'}$ describes the interannual covariance of monthly relative frequency and monthly latitude ($p'$ and lat$'$). For a given year, $\delta_{P'Lat'}$ is positive if monthly $p'$ and lat$'$ for that year are positively correlated and negative if they are negatively correlated. A conceptually similar decomposition method has been used to quantify the effect of inter-basin changes in TC frequency on the global poleward shift of TC tracks[21].

Tables 1 and 2 provide the regressed linear trends of the three contributions to the annual-mean latitude for various TC metrics in the Best Track ensemble mean and ERA-Interim TC datasets. Figure 3 and Supplementary Figs. 5 and 6 illustrate the timeseries of the three contributions for all-track-points, genesis and LMI,

respectively. In the satellite era, among the three contributors, $\delta_{P'}$ has the largest and most significant poleward trends, with smaller and less significant trends in $\delta_{P'Lat'}$ and $\delta_{Lat'}$. For example, in the Best Track ensemble mean, for all-track-points, annual $\delta_{P'}$ has a significant upward trend of 43±19 km/decade, while annual $\delta_{Lat'}$ and $\delta_{P'Lat'}$ have no significant trends (26±30 and 9±19 km/decade, respectively; both not significant at 95% confidence). The generally insignificant trends in annual $\delta_{Lat'}$ of TC metrics are related to the large interannual variability in these metrics. The northward migration of annual $\delta_{P'Lat'}$ is not statistically detectable either, except for all-track-points in ERA-Interim. These identifiable trends in annual $\delta_{P'Lat'}$ in ERA-Interim are due to an increased covariance of monthly relative frequency and monthly latitude, suggesting that the changes with time in monthly relative frequency and latitude may not be independent. For most of the TC metrics, $\delta_{P'}$ alone accounts for 40–60% of the total poleward migration rates. Insignificant trends in annual $\delta_{Lat'}$ are consistent in all TC metrics and in all Best Track sources (Supplementary Table 1), even though the trends are mostly northward. In contrast, significant northward trends in annual $\delta_{P'}$ in all TC metrics and in all data sources imply small uncertainty in trends. Annual $\delta_{P'Lat'}$ has detectable trends in at least one Best Track source, but the trends are much smaller than those of annual $\delta_{P'}$. Thus, we conclude that over the satellite era the northward migration of WNP TCs is predominantly caused by changes in the TC seasonality ($\delta_{P'}$).

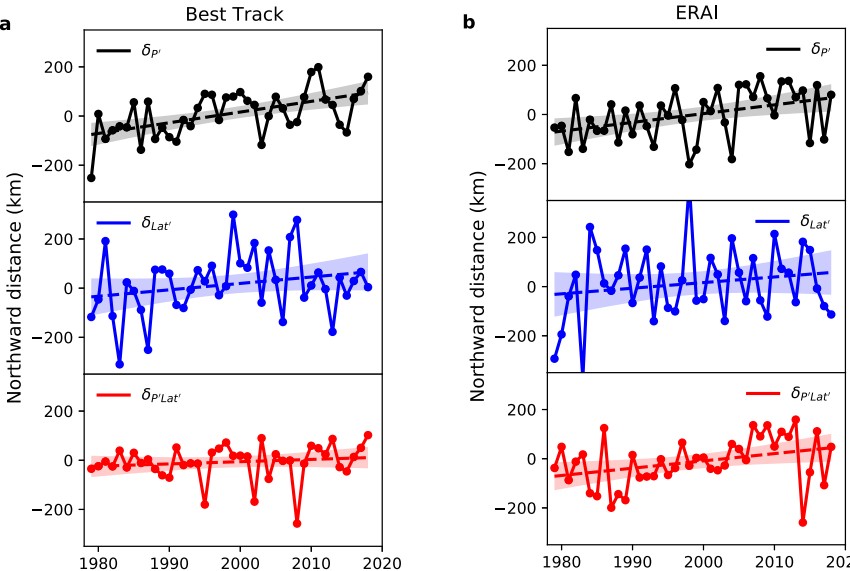

**Fig. 3 Changes in three seasonality components of annual-mean latitude of tropical cyclone all-track-points. a** Decompositions of annual-mean latitude of all-track-points by three seasonality contributing components, representing the effects of departures (from the respective time means) of monthly relative frequency ($\delta_{P'}$, black), monthly latitude ($\delta_{Lat'}$, blue) and the covariance of the two terms ($\delta_{P'Lat'}$, red), respectively, over 1979–2018, in the Best Track ensemble mean. **b** as **a**, but in the ERA-Interim TC dataset. Dashed lines represent the linear trend, with shading showing 95% confidence interval for the linear fit; the El Niño Southern Oscillation and Pacific Decadal Oscillation effects are removed by a multivariate regression; latitude is converted to distance from the equator (km).

Changes in recurvature of storms could affect the poleward migration of TC tracks. In the WNP, recurving TCs normally form and travel further north than straight-moving TCs (Supplementary Fig. 7). An increasing proportion of recurving TCs might result in a poleward trend in average TC location. However, in both datasets, over 1979–2018, we find no significant changes in the proportions of either recurving TCs or straight-moving TCs, suggesting no significant recurvature changes in WNP TCs. Instead, we find that straight-moving TCs become significantly fewer in late season, contrasting with no significant changes in recurving TC frequency across all seasons (Supplementary Fig. 8). Thus, for straight-moving TCs, the peak-season relative frequency has significantly increased, and the late-season relative frequency has significantly reduced. Because of the larger changes in straight-moving TC seasonality ($\delta_{P'}$), these storms tend to have larger migration rates in annual-mean location than recurving TCs (Supplementary Tables 2 and 3). This finding again suggests the importance of TC seasonality change in the poleward migration of WNP TCs.

**Migration of the annual-mean position of TCs over the pre-satellite era.** We extend the trend analysis back to 1951 using Best Track data (Fig. 4, Table 1 and Supplementary Table 1). Caution must be taken with these statistical results, because of the low reliability of Best Track data in the pre-satellite era[22]. In the early period (1951–1978), a significant northward migration is only found in genesis, largely attributable to changes in annual $\delta_{Lat'}$ (Table 1 and Supplementary Fig. 9). This northward change is found both in JTWC and CMA data (Supplementary Table 1). However, the trend in annual $\delta_{Lat'}$ is likely contaminated by the transition of intensity estimates from aircraft to satellite reconnaissance in the basin during the 1970s and the 1980s[11]. Dvorak techniques for intensity estimation from satellite reconnaissance can significantly underestimate storm intensity for weak stages of the storm lifecycle or for weak storms[11–13]. As a result, a fixed threshold of intensity to define genesis (here, a threshold maximum sustained wind ≥34 knots) would yield aircraft-based genesis locations further equatorward

than satellite-based genesis locations. We speculate that the gradual decline of aircraft reconnaissance in observations[11] can lead to a spurious northward migration of genesis. In the early period, the net latitudinal transit distances have significant decreasing trends, driven by the changes in $\delta_{Lat'}$ as well (Table 1 and Supplementary Table 1). Again, these trends in $\delta_{Lat'}$ are likely caused by observation bias in the Best Track data. First, in the transition period discussed above, aircraft reconnaissance may lead to a more equatorward genesis location, and hence a longer storm lifetime and a larger meridional travelling distance. Secondly, there is spatial inhomogeneity in observed tracks over the pre-satellite era. In earlier years, the frequency and track points of TCs over the sea are likely underestimated[22,23]. These missed TCs or track points are more likely to be at lower latitudes, since TCs over the sea are typically at lower latitudes than those over land. Both factors could artificially reduce the latitudinal transit distance of TCs in the early period, as more satellite-based and lower-latitude TCs are gradually counted.

Here, one key message is that even in the questionable period of the pre-satellite era, there is still a noticeable poleward migration caused by changes in TC seasonality ($\delta_{P'}$) in TC metrics (all-track-points, genesis, LMI and lysis) (Fig. 4 and Supplementary Figs. 9 and 10, Table 1 and Supplementary Table 1). The significance of migration trends in $\delta_{P'}$ is slightly lower in the pre-satellite (at 90% confidence) than in the satellite era (at 95% confidence), perhaps due to the shorter time period. The signs and magnitudes of migrating trends in $\delta_{P'}$ are comparable between the two periods. However, $\delta_{P'}$ may be affected by the temporal and spatial inhomogeneity in Best Track observations, particularly if the seasonal distribution of undetected TCs (mentioned above) differs from the seasonal distribution of detected TCs. Thus, statistically, the changes in annual $\delta_{P'}$ are crucial for the poleward migration of WNP TCs in both the pre-satellite era and satellite era.

**Seasonally non-uniform effect of changes in the environmental conditions.** It is important to understand why the late-season TC frequency declines when the peak-season frequency is nearly

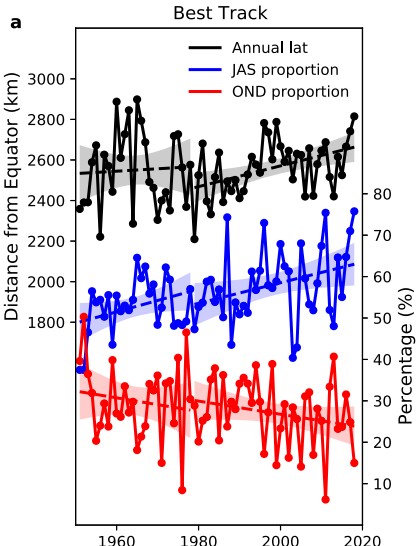
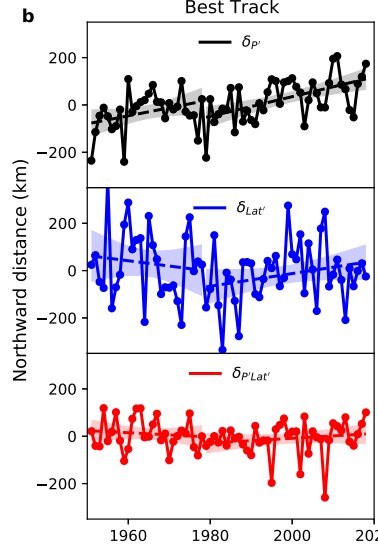

**Fig. 4 Changes in annual-mean latitude of tropical cyclone all-track-points, and changes in three seasonality components, over the extended period 1951–2018. a** Timeseries of annual-mean latitude of all-track-points (black), and timeseries of peak-season (July–September; JAS, blue) and late-season (October–December; OND, red) relative frequencies of all-track-points in the year, in two epochs (the pre-satellite era 1951–1978 and the satellite era 1979–2018) of the whole period 1951–2018, in the Best Track ensemble mean. **b** Decompositions of annual-mean latitude of all-track-points by three seasonality contributing components, representing the effects of departures (from the respective time means) of monthly relative frequency ($\delta_{P'}$, black), monthly latitude ($\delta_{Lat'}$, blue) and the covariance of the two terms ($\delta_{P'Lat'}$, red), respectively, in two epochs of the whole period 1951–2018 in the Best Track ensemble mean. Dashed lines represent the linear trend, with shading showing 95% confidence interval for the linear fit; the El Niño Southern Oscillation and Pacific Decadal Oscillation effects are removed by a multivariate regression; latitude is converted to distance from the equator (km).

unchanged, as this is key to the increase in peak-season relative frequency and the apparent WNP TC poleward migration. We focus on the satellite era when the data are more reliable. In the late season, the main development region (MDR) of WNP TCs is in the central tropical Pacific (140°E–160°W and 0–20°N; red box in Fig. 5a–c); in the peak season, the MDR shifts further northwest to the western tropical Pacific (110–170°E and 5–25°N; red box in Supplementary Fig. 11a–c). Usually, at least 75% of WNP TCs form in the MDR (Supplementary Fig. 12), except for the late season in Best Track (accounting for 51% of WNP TCs). This lower percentage is because in the late season the Best Track-defined genesis is further west, while the ERA-Interim genesis defined by our tracking method is relatively further east. We find that the trend and variability of WNP TCs are dominated by TCs formed in the MDR, for both seasons and in both datasets. In the past four decades, observations show a La Niña-like trend pattern in sea surface temperatures (SSTs) and other TC-related environmental factors[24–27]. These environmental changes have been linked to the recent TC changes for individual seasons[28,29].

Next, we further highlight the seasonally non-uniform effect of these environmental changes on WNP TC frequency under a changing climate, to elucidate the change in TC seasonality. The environment data are from ERA-Interim. Over the past four decades, after the linear effects of ENSO and PDO have been removed, the late-season relative SSTs (relative to the average of global SSTs in 30°S–30°N) have an enhanced west-to-east SST gradient across the equatorial Pacific (Fig. 5a). As a thermodynamic response, the mean sea level pressure decreases significantly in the ascent region of the western Pacific and increases in the eastern Pacific (Fig. 5b). The vertical wind shear (VWS; the difference of wind flows at 200 and 850 hPa), a crucial factor for TC activity, increases significantly in the central tropical Pacific, with a westward tilt with latitude (Fig. 5c), which resembles the late-season MDR. This VWS increase is driven by the local strengthening of low-level easterlies (trade winds) and

upper-level westerlies (Fig. 5d). A stronger Pacific Walker Circulation, with descent in the eastern Pacific and ascent in the western Pacific (Fig. 5e), is embedded in a large-scale strengthening of zonal overturning circulation in the tropical Pacific (Fig. 5d), including the MDR. The Pacific Walker Circulation change in turn relates to the increased west-to-east SST gradient in the tropical Pacific. Figure 6 shows the timeseries of MDR-averaged environments, against the MDR genesis frequency, for the late season. MDR-averaged SSTs have a significant trend (0.10±0.05 °C/decade), while there is no significant trend in relative SSTs. Both relative vorticity at 850 hPa ($-0.24 \pm 0.22 \ 10^{-6} \ s^{-1}$/decade) and VWS (0.67±0.39 m/s/decade) have significant trends, associated with a significant trend in 850-hPa zonal wind velocity ($-0.34\pm0.2$ m/s/decade). Related to this, the MDR genesis from ERA-Interim is strongly anti-correlated with VWS ($r = -0.50$) and correlated with 850-hPa zonal wind velocity ($r = 0.50$), but less correlated with SSTs or relative SSTs ($r = 0.38$). The unfavourable atmospheric conditions in the MDR, related to the stronger Pacific Walker Circulation, are likely responsible for the decline of late-season TC genesis.

Equally, we evaluate the trends in the peak-season environmental conditions (Supplementary Fig. 11). The trends have a regional pattern like that for the late season, but with smaller values. The Pacific Walker Circulation is also enhanced in the late season, but the main changes are either in the central and eastern tropical Pacific (160°E–140°W) or in the middle-to-upper levels (600–200-hPa). These changes have little impact on the ambient environmental conditions for most peak-season TCs. The changes in MDR-averaged environmental variables are weak (Supplementary Fig. 13), with small trends in relative SSTs (0.03 ± 0.03 °C/decade) and VWS (0.18±0.18 m/s/decade), which are statistically significant only at 90% confidence, and with no significant trends in other variables. The small changes in local environmental conditions correspond to no significant trend in peak-season TC frequency.

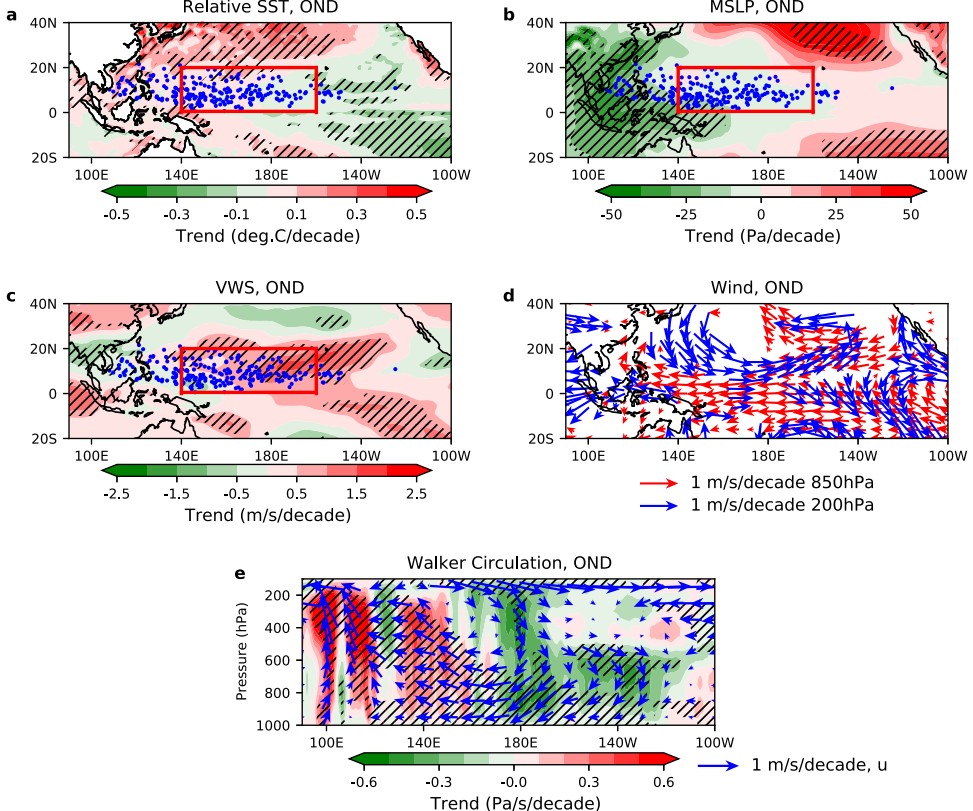

**Fig. 5 Changes in late-season environmental conditions in the Pacific. a–d** Linear trends (shading) of relative sea surface temperatures (relative SSTs), mean sea level pressure (MSLP), vertical wind shear (VWS) and wind velocity at 850 and 200 hPa (vectors), for the late season (October–December; OND), over 1979–2018, in ERA-Interim. Relative SSTs are SSTs minus the average of global SSTs in 30°S–30°N. Blue dots present late-season TC genesis in the ERA-Interim TC dataset; red box (140°E–160°W and 0–20°N) represents the main development region (MDR) of late-season TCs. **e** Linear trends of the late-season Walker Circulation (vectors) and vertical velocity (shading; positive values for uplifting) in the zonal-vertical section averaged over 5°S–5°N, over 1979–2018, in ERA-Interim. Hatched areas in **a–c, e** and vectors in **d** show 95% confidence for the linear fit; the El Niño Southern Oscillation and Pacific Decadal Oscillation effects are removed by a multivariate regression; in **e**, trends of vertical velocity are multiplied by 10$^2$ for display.

This study has shown that the average location of WNP TC tracks has systematically shifted northward, shown by TCs both forming and moving further north, especially in the satellite era when the track data are more reliable. The northward migration is related to significant changes in the seasonality of TC frequency, associated with relatively more peak-season TCs and fewer late-season TCs. When evaluated across all seasons, there is no identifiable trend in the seasonal latitude. Our findings offer a different interpretation of the northward migration of WNP TCs, by decomposing the changes into several seasonal components, which might all be driven by anthropogenic forcing but likely through different mechanisms.

## Discussion

The Pacific Walker Circulation change over the last four decades has been associated with the long-term natural climate variability (e.g., the Interdecadal Pacific Oscillation and the Atlantic Multi-decadal Oscillation)[27,30,31]. However, it remains controversial how much of these observed changes are attributable to anthropogenic forcing, partially because in attribution studies contemporary GCMs disagree on the spatial pattern of Pacific SST response to anthropogenic forcing[24,26,32,33]. Our analysis indicates that changes in the large-scale environments have a seasonally non-uniform effect on WNP TC frequency, i.e., by significantly reducing late-season TCs and barely affecting peak-season TCs. This inconsistent effect is hypothesised to be

responsible for the poleward migration of the annual-mean location of WNP TCs. A clearer understanding of the fundamental causes of observed changes in the large-scale environments in the WNP basin, and whether these changes will hold in the future, would further benefit understanding and predictions of TC seasonality changes.

Over the last few decades, the effect of the meridional shift of climatic conditions on the WNP TC migration may be small and statistically undetectable, or the meridional shift itself due to anthropogenic forcing may be small[34]. Instead, like we find here, the zonally asymmetric changes in the tropical Pacific environment, associated with a stronger Pacific Walker Circulation, could play an important role in the northward migration of WNP TCs through changing the TC seasonality. The same analysis could be applied to the changes in other TC activity measures, including the annual means of longitude[35], translation speed[36,37] and intensity[38,39], where these measures also exhibit distinct seasonal variations. Such studies will improve our understanding of the links between climate change and long-term TC activity trends, including TC-related hazard exposure and risk. It may also help scientists to bridge gaps between simulated and observed trends in TC properties, by evaluating and improving the ability of GCMs to simulate regional-scale environmental changes related to TC seasonality under a changing climate. This will increase our confidence in using climate models to project future changes in TC activity both globally and regionally.

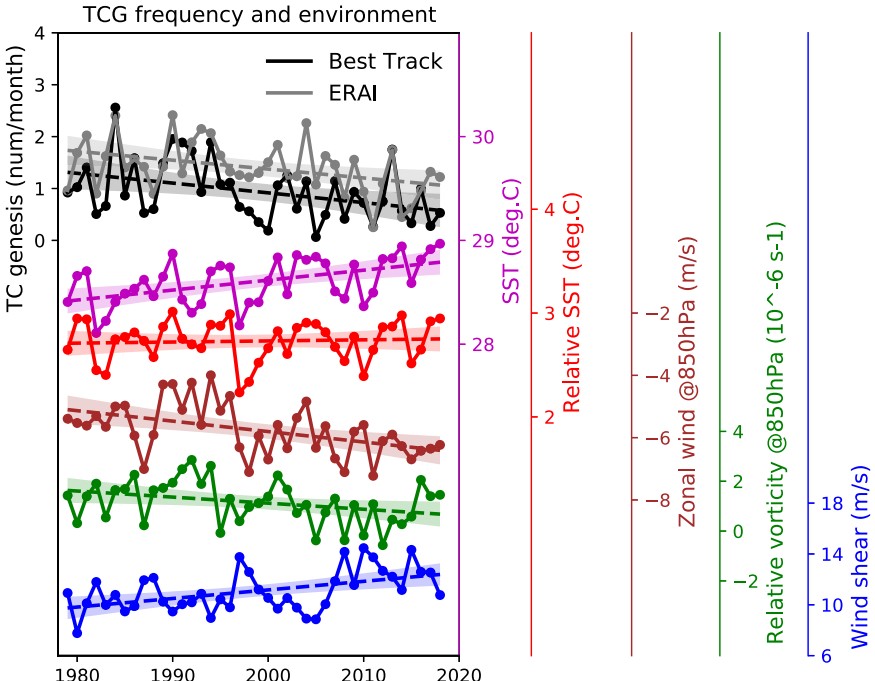

**Fig. 6 Changes in late-season tropical cyclone frequency and environmental conditions in the main development region.** Timeseries of monthly frequency of TC genesis (TCG) developed in the main development region (MDR, red box in Fig. 5), for the late season (October–December; OND) over 1979–2018, in the Best Track ensemble mean (black) and ERA-Interim (grey) TC datasets. Timeseries of sea surface temperatures (SSTs, magenta), relative SSTs (red), 850-hPa zonal wind (brown), 850-hPa relative vorticity (green) and vertical wind shear (blue), averaged over the MDR for the late season over 1979–2018, in ERA-Interim. Relative SSTs are SSTs minus the average of global SSTs in 30°S–30°N. Dashed lines represent the linear trend, with shading showing 95% confidence interval for the linear fit; the El Niño Southern Oscillation and Pacific Decadal Oscillation effects are removed by a multivariate regression.

## Methods

**Best Track data**. TC Best Track observations from three national meteorological agencies—the Joint Typhoon Warning Centre (JTWC), the Japan Meteorological Agency (JMA) and the China Meteorological Administration (CMA)—for 1951–2018 are used to synthesise a multi-source ensemble (Best Track ensemble). Best Track observations include the location and intensity (maximum sustained wind speed and minimum sea level pressure) at 6-hourly intervals. As TC track uncertainty is large for weak storms[12], our analysis of Best Track data only considers TCs that reach severe tropical storm intensity (i.e., maximum sustained winds ≥48 knots or ≥24 m/s).

Our Best Track ensemble consists of various TC location metrics (see *TC location metrics* in the section for details). These TC location metrics are derived for each storm, in every Best Track source, separately, before synthesising the ensemble. The spread of the ensemble reflects inter-agency uncertainty in TC estimates. Maximum sustained winds are estimated differently at the three agencies: JTWC uses 1-min mean of maximum sustained winds, while CMA and JMA use 2- and 10-min means, respectively. There is no simple conversion between these wind speeds[14]. The extratropical cyclone stage is included in Best Track, but the years when this began vary among agencies[16].

In our analysis, we divide the Best Track records into the pre-satellite (1951–1978) and satellite (1979–2018) eras. Caution should be taken with the statistical results for the pre-satellite era. In the pre-satellite era, Best Track data have low reliability due to observation biases[11,22,23] (see the main text). In the pre-satellite era, the JMA Best Track data usually have missing TC intensity estimates, although the categories of storm are provided. This makes it impossible to calculate the location metric of LMI and other related metrics (transit distances of developing and dissipating lifecycle phases). Thus, the Best Track ensemble for these metrics in the pre-satellite era only includes the JTWC and CMA data.

**ERA-Interim TC track data**. TCs are identified and tracked in the ECMWF Interim Reanalysis (ERA-Interim)[40] from six-hourly atmospheric data, during 1979–2018, using the method described in refs. [20,41] (see references therein for more details). First, the vertical average of the relative vorticity between 850 and 600 hPa is obtained. This is then spatially filtered using spherical harmonics to T63 resolution; the large-scale background with total wavenumbers $n \le 5$ is removed. Vorticity maxima (in the Northern Hemisphere) are determined on the T63 grid and then used as starting points to obtain the off-grid locations using B-spline interpolation and maximisation methods. In the first instance, all positive vorticity centres that exceed $0.5 \times 10^{-5} \, s^{-1}$ in the range 0°–60°N are identified through the

data timeseries. The tracking is performed by first initialising a set of tracks using a nearest-neighbour method and then refining them by minimising a cost function for track smoothness subject to adaptive constraints on track smoothness and displacement distance in a time step.

After the tracking is complete, the full T63 vorticity maxima at levels from 850 hPa up to 200 hPa (850, 700, 600, 500, 400, 300 and 200 hPa) are added to the tracks using a recursive search within a 5° radius (geodesic) of the tracked centre. This is used to test for the existence of a coherent vertical structure and a warm core. Other variables are also added to the tracks, including the maximum 10 m wind speed. The 10 m wind speed maxima are identified by searching within a 6° geodesic radius centred on the TC track points.

TCs are selected from all tracks using the following selection criteria, which produce the most coherent tracks and retains the full lifecycles of the systems, including their pre-TC and post-TC stages:

1. the T63 relative vorticity at 850 hPa must attain a threshold of at least $6 \times 10^{-5} \, s^{-1}$;
2. the difference in vorticity between 850 and 200 hPa (at T63 resolution) must be greater than $6 \times 10^{-5} \, s^{-1}$ to provide evidence of a warm core;
3. the T63 vorticity centre must exist at each level between 850 and 200 hPa for a coherent vertical structure; and
4. criteria 1 to 3 must be jointly attained for a minimum of four consecutive time steps (one day) and only apply over the oceans.

We analyse only ERA-Interim TC tracks that originate in 0–25°N and enter the WNP basin (0–60°N, 100–180°E), as our focus is on the WNP TCs from the deep tropics. TC intensity is measured by maximum 10 m wind speed. A criterion of LMI ≥ 15 m/s is set to filter out the weak tropical storms, which have the most uncertainty in identification[20]. The criterion of LMI ≥ 15 m/s is chosen based on the ERA-Interim spatial resolution, following ref. [42].

A few points are worth noting about the ERA-Interim TC dataset. First, compared to Best Track, the ERA-Interim storms have an extended lifecycle[20], allowing analysis of changes over the full system lifetime, including the pre-TC and post-TC stages, over which Best Track has large inter-agency uncertainty in recording location and intensity[12,14,16]. The earlier stage of TC generation is more associated with the genesis-related environment than the later stage of storms, giving the ERA-Interim TC dataset an advantage in interpreting the environmental effects on TC genesis. Secondly, apart from the different tracking method, the ERA-Interim TC dataset is distinct from Best Track because TC observations are not assimilated into ERA-Interim. Thirdly, a caveat of ERA-Interim dataset is a systematic underestimate of climatological TC intensity, which is associated with a

northward displacement of LMI position, related in part to the low resolution of the atmospheric model used in the reanalysis[20]. Because ERA-Interim reproduces well the WNP TC seasonality, the dataset is used to evaluate the migration trends in TC metrics associated with the seasonality change. Please note that in our analysis, we do not use the ERA-Interim TC dataset in preference to Best Track. Instead, we use ERA-Interim as supporting data to examine the robustness of the results initially derived from Best Track.

**TC location metrics.** We use four metrics to represent absolute locations of each TC, including all six-hourly track points (all-track-points), tropical cyclogenesis (TC genesis), LMI and cyclolysis (TC lysis). The LMI location is determined by the first track point where TC reaches its LMI. In Best Track, the TC genesis location is determined by the first track point where TC intensity reaches 34 knots (or 18 m/s, or category of tropical storm, where applicable), while the TC lysis location is the last track point at any intensity. The extratropical cyclone stage is included in Best Track data, but the years when this began vary among agencies[16]. This would lead to uncertainty in TC lysis location and other related metrics in the Best Track ensemble. ERA-Interim TCs have different definitions for genesis and lysis due to the different identification scheme. In ERA-Interim, TC genesis and lysis are the first and last points of the identified TC track, for which relative vorticity exceeds the given thresholds (see ERA-Interim TC track data above for details).

We use another four metrics to measure various TC net latitudinal transit distances. For each TC, the net track transit distance is the latitudinal difference between lysis and genesis locations; the net transit distance of the storm developing phase is the latitudinal difference between LMI and genesis locations; the net transit distance of the storm dissipating phase is the latitudinal difference between lysis and LMI locations; and the net transit distances of all-track-points are calculated as the latitudinal differences from all-track-points to the genesis location. Note that in the pre-satellite era (1951–1978), because the JMA Best Track data do not usually record TC intensity estimates, the location of LMI and other related metrics (transit distances of the storm developing and dissipating phases) are not calculated. For these three metrics, only the JTWC and CMA data are used in the Best Track ensemble.

We also estimate the location metrics for recurving and straight-moving TCs. Recurving and straight-moving TCs are identified as suggested in ref. [43]. Recurving TCs have the following selection criteria: (1) storm recurves east at the westernmost point over the full lifetime; (2) the subsequent track point is further north than the recurving point. Straight-moving TCs are the storms that do not simultaneously meet above two criteria.

**Decomposing the annual-mean latitude of TC location metrics.** The annual-mean latitude of each location metric, Lat($i$), in year $i$ with $N$ samples, can be simply written as

$$\text{Lat}(i) = \sum_{n=1}^{N} \text{lat}(n)/N \tag{1}$$

where lat($n$) is the latitude for each sample $n$.

By further classifying the $N$ samples into 12 calendar months ($m$), Eq. (1) can be rewritten on a monthly bases as

$$\text{Lat}(i) = \sum_{m=1}^{12} \text{lat}(m,i) * p(m,i) \tag{2}$$

where lat($m,i$) and $p(m,i)$ are the monthly latitude and monthly relative frequency of TC samples for the calendar month $m$ in year $i$. Assuming the sample size for the calendar month $m$ is $N_m$ ($\sum_{m=1}^{12} N_m = N$), lat($m,i$) and $p(m,i)$ can be specified as

$$\text{lat}(m,i) = \sum_{n=1}^{N_m} \text{lat}(n)/N_m$$
$$p(m,i) = \frac{N_m}{N} \tag{3}$$

lat($m,i$) and $p(m,i)$ can be decomposed into long-term time means ($\overline{\text{lat}(m)}$ and $\overline{p(m)}$) and anomaly terms departing from the time means in year $i$ (lat$'(m,i)$ and $p'(m,i)$). Finally, annual Lat($i$) is expressed as Eq. (4):

$$\begin{aligned}
\text{Lat}(i) &= \sum_{m=1}^{12} (\overline{\text{lat}(m)} + \text{lat}'(m,i)) * (\overline{p(m)} + p'(m,i)) \\
&= \underbrace{\sum_{m=1}^{12} \overline{\text{lat}(m)} * \overline{p(m)}}_{\text{climatic term}} + \underbrace{\sum_{m=1}^{12} \overline{\text{lat}(m)} * p'(m,i)}_{\text{proportion term}} + \underbrace{\sum_{m=1}^{12} \text{lat}'(m,i) * \overline{p(m)}}_{\text{latitude term}} \\
&\quad + \underbrace{\sum_{m=1}^{12} \text{lat}'(m,i) * p'(m,i)}_{\text{covariance term}} \\
&= \bar{\delta} + \delta_{P'} + \delta_{\text{Lat}'} + \delta_{P'\text{Lat}'}
\end{aligned} \tag{4}$$

The deviation of annual Lat($i$) from the long-term climatic average ($\bar{\delta}$) consists of the three terms, denoted as $\delta_{P'}$, $\delta_{\text{Lat}'}$ and $\delta_{P'\text{Lat}'}$, which quantify the effects of the interannual departures of monthly relative frequency ($p'$) and monthly latitude (lat$'$) from their time means, and the interannual covariance of the 2 monthly anomalies ($p'$ and lat$'$), respectively.

**Statistical analyses.** For the trend analysis, we estimate the trend (denoted as $b$) using a linear least-squares regression, and we also estimate the error bars (denoted as err) of the trend by a two-tailed 95% confidence interval under the assumption that the residuals of the regression follow a normal distribution. $b \pm$ err represents the 95% confidence estimate of the trend value. The trend is tested for statistical significance for a null hypothesis that the trend is zero (i.e., a significant trend means that the interval of trend ($b \pm$ err) does not include zero). The Pearson correlation coefficient (denoted as $r$) is used to measure the correlation between the timeseries of two variables. A two-tailed $t$-test with a $p$ value of 0.05 is used to test significance, with a null hypothesis of a zero correlation. In other words, in our paper, the statistically significant values ($b$ or $r$) are at the 95% confidence level, unless stated otherwise.

Before the above statistical analyses, the effects of ENSO and PDO are removed from the timeseries of TC metrics and environmental variables (e.g., SST, wind velocity), as suggested in ref. [2]. To do so, a multiple linear regression on the yearly ENSO and PDO indices with a linear least-squares method is first estimated for the timeseries of data. Yearly ENSO and PDO indices are averages of the monthly indices over the typhoon season July–November. The values determined by ENSO and PDO indices are then removed from the timeseries of TC or environmental data to obtain the residuals. The residuals will feed into the above trend and correlation analyses. These processes are done for the timeseries of both yearly and monthly values.

## Data availability

Best Track data are obtained from three national meteorological agencies, which are the Joint Typhoon Warning Centre (JTWC, https://www.metoc.navy.mil/jtwc/jtwc.html?western-pacific), the Japan Meteorological Agency (JMA; www.jma.go.jp/jma/jma-eng/jma-center/rsmc-hp-pub-eg/trackarchives.html) and the China Meteorological Administration (CMA; http://tcdata.typhoon.org.cn/en/zjljsjj_sm.html). Monthly ENSO and PDO indices are retrieved from NOAA's National Centres for Environmental Information (www.ncdc.noaa.gov/teleconnections/). The ERA-Interim climate reanalysis[40] was generated and distributed by ECMWF (https://www.ecmwf.int/en/forecasts/datasets/reanalysis-datasets/era-interim). Computing and data storage facilities were provided by JASMIN (https://jasmin.ac.uk).

## Code availability

The code and scripts used to analyse the data and to generate the plots in this paper are available from the corresponding author on request.

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

## Acknowledgements

X.F. and K.I.H. were supported by the Met Office Climate Science for Service Partnership for China and the Weather and Climate Science for Service Partnership for Southeast Asia, as part of the Newton Fund. N.P.K. was supported by an Independent Research Fellowship from the Natural Environment Research Council (NE/L010976/1) and by the Global Challenges Research Fund, via Atmospheric hazards in developing Countries: Risk assessment and Early Warning (ACREW; NE/R000034/1). We thank Professor Theodore G. Shepherd at the University of Reading for comments on the earlier version of the manuscript.

## Author contributions

X.F. designed and performed the research; K.I.H. tracked the storm tracks in the climate reanalysis; X.F., N.P.K and K.I.H. discussed the results and wrote the manuscript.

## Competing interests

The authors declare no competing interests.
