## [Peer Review File · Nature Communications]

REVIEWER COMMENTS

Reviewer #1 (Remarks to the Author):

Review of "Poleward migration of western North Pacific tropical cyclones related to changes in cyclone seasonality" for Nature Communications manuscript NCOMMS-20-46326-T

Recommendation

The authors analyze the poleward shift of typhoons using Best Track data from Japan with further support from objectively tracked reanalysis cyclone tracks. While this trend has garnered attention lately as a signal for a connection to climate change via an expanding Hadley cell circulation. The authors decompose the trend into two seasonal portions – summer (JAS) and autumn (OND) – showing that while the summer portion has undergone some change, the autumn portion has experienced a significant decrease in relative numbers leading to the noted shift in latitude. The details are interesting and tell a bit more complex exists in the poleward shift. Given the recent interest in this topic, the rigor of the analysis and the results, I would recommend acceptance with minor revision.

Specific comments

1. Title – Since tropical cyclones in this region are only called typhoons, the title could be more succinctly as "Poleward migration of typhoons related to changes in cyclone seasonality"
2. Abstract – is clear and well written
3. Line 74 – perhaps cyclolysis vice lysis?
4. Line 95 – the series of numbers are less clear in this sentence ... might be easier to read if written as " ... LMI (42+-31), and cyclogenesis (38+-31) migrated ... all track points (70+-35) .." [where I write +- for plus or minus ... not sure where that is on my keyboard).
5. Line 106 – The statement that ERA Interim trends are larger is confusing because I first looked at the lower portion of Table 1. ... I think it would be better to separate Table 1 into 2 tables. One for each of the different periods of record, for clarity.
6. Paragraph at line 146 – I must object to this use of parentheses to denote opposites. As described in AGU Eos , they do not add to readability, in fact, more the opposite. Please rewrite this paragraph. Specifically, they state: "Making a journal article a few words longer for clarity and to avoid confusion is well worth the expense in extra bits of storage." Why? I am forced to read the paragraph twice and mentally note that the parentheses at lines 146, 151, 156 and 160 (second set) are examples or figures while the parentheses at lines 147, 149, 151 (2nd group), 152, 153, and 160 (1st set) are opposites. Such intermingling would cause syntax errors were this programming ... and to that extent, readers are not programs that can mentally carry the argument of all the opposites along. Another sentence or two would make your points clearer than the seemingly efficient usage of parentheses.
<https://eos.org/opinions/parentheses-are-not-for-references-and-clarification-saving-space>
7. Line 171 ... same line has multiple uses for parentheses. For instance, the sentence could be written "For a given year, dP_{Lat} is proportional to the correlation of p' and lat' (that is, positive for same-signed variations in p and lat). .. Assuming "same signed variations" is akin to correlation.
8. For overall readability, please consider replacing "JAS" with "summer" and "OND" with "autumn" ... these phrases are used in introducing the topic on lines 120 and 122 ... but the author seems to prefer JAS and OND. While the acronym is specific, the words are more readable and clearly defined in lines 120/122 so why not use them.
9. Line 415 – should be "Values in parentheses..."
10. Figure 1a – would be clearer if the left hand y axis only extended down to 2000km (since the red & blue lines correspond to the proportion on the right hand axis). Likewise, the proportion axis could stop at 80%. Other figures in the (and in supplement) could likewise be adjusted for clarity. [Fig 5c has a nice demonstration of this.]
11. Figure 5a – Put the MDR in this plot as well.

Reviewer #2 (Remarks to the Author):

Review of "poleward migration of western North Pacific tropical cyclones related to changes in cyclone seasonality"

GENERAL COMMENTS

1. This is an interesting paper to extend previous studies on poleward migration of tropical cyclones for different seasons. While the idea is good and innovative, it is unclear as to how the results are obtained because of the following:
 - a. lack of the definition of the metrics (Specific Comments 1, 9 and 10), and
 - b. the use of the ERA-Interim dataset as the primary source of data when this dataset is well known for many uncertainties – see Specific Comment 4.
2. With the problems associated with the ERA-I dataset, why is the best-track dataset from JMA or JTWC not used as the primary source of data for the study?
3. Throughout the manuscript, no result of statistical test of any of the trends was provided except in Table 1. It is therefore not clear which of the trends are statistically significant.
4. Normally, these deficiencies, together with those mentioned below in the Specific Comments, would lead to rejection of the manuscript. However, given the innovative idea on a possible seasonality of poleward migration, and hence how such seasonality might contribute to the overall migration trend, if any, it is recommended that the authors address all these issues mentioned in this review, re-do the analyses using the JTWC or JMA dataset, compare the results in the pre- vs post-satellite eras, and then re-submit the manuscript.

SPECIFIC COMMENTS

1. Lines 73-74:
 - (a) Throughout the manuscript, there is no definition of cyclogenesis, LMI and lysis. For example, is genesis defined as the first point of the TC reaching 25 or 30 kt? For LMI, in many situations, the LMI can be maintained for a period of time, and in some cases, a TC can weaken briefly after reaching the LMI and then re-intensify to the LMI again. So which position is defined as the latitude of LMI? Similarly, is the lysis position the last position before dissipation irrespective of intensity? What about cases of extratropical transition? Many of the TCs dissipate over land but not over the ocean. Are land points considered? All these definitions and how the data points are obtained must be clearly presented and explained so that the reader can compare the results with those from previous studies.
 - (b) The net latitudinal transit distances need to be clearly explained. Recurring TCs have shorter transit distances, and hence the net latitudinal displacement could be less.
2. Line 78: Please explain how to have >800 samples per year.
3. Line 81: Data from 1951 to the pre-geostationary satellite era (approximately 1975) have been criticized as having many problems, especially for intensity, which implies large uncertainty in the determination of the position of LMI. This problem therefore has to be discussed as to how it is taken into consideration in the subsequent analyses and evaluation. Data in the post-satellite era should be studied independently to see if the conclusions are different.
4. Lines 81-82: Estimating the metrics mentioned in lines 72-73 using ERA-Interim has a number of problems as well. This dataset does not have the horizontal resolution to give a good estimate of intensity, and hence the latitude of LMI determined from the dataset is likely incorrect. Figure 2c and Extended Data Figure 3c show consistent differences between the best track data and the ERA-I dataset, which suggests that the latter has some inherent problem in providing a good estimate of LMI and its latitude. Further, this dataset tends to give more vortices than the observed, especially near the dateline. This can actually be seen in Fig. 2a, Extended Data Figure 3a, and the location of the black dots in Fig. 5 of the manuscript. Thus, while ERA-I could be used to provide some estimate of the metrics, discussions of the results from this dataset must include caveats on these uncertainties. In fact, it is not clear why the authors choose this dataset over the

best-track dataset.

5. Lines 82-83: What is meant by “an extended lifecycle”? Why is the best-track dataset not able to provide the same information?
6. Lines 86-87: Please explain how ENSO and PDO are actually removed using multiple linear regression for each of the metrics examined.
7. Lines 88-89: It is doubtful that the conclusions can be “identical”. No difference at all?
8. Line 97: What is meant by “due to small trends in the 2010s”?
9. Line 135: “TC lifetime”? A new metric? What is the definition?
10. Lines 381-383: What is meant by “monthly latitude”? Please also give an equation for the definition of $p(m, i)$. In fact, it is not clear how Equation (1) can be rewritten to Equation (2). Please explain clearly why this can be done.
11. Table 1: Why are there no entries for LMI for the best-track dataset?

Reviewer 3 (Remarks to the Author):

This paper claimed that the shift of average location of observed western North Pacific (WNP) tropical cyclones (TCs) over the last four decades is associated with relatively more (less) frequent TC genesis in summer (autumn). The authors also concluded that the TC frequency decline in autumn is attributed to the increase in vertical wind shear in the central tropical Pacific. This paper has a merit in that it offers a different perspective on long-term trends in TC activity, by decomposing the annual-average statistics into seasonal components (as authors mentioned). However, unfortunately there are some issues that need to be resolved before this manuscript can be acceptable for publication, as summarized below.

Major concerns

- 1) The authors claimed that “(line 7-8) While anthropogenic forcing is hypothesized to be the cause, climate-model simulations do not reproduce this trend in response to anthropogenic forcing” and “(line 32-34) However, for WNP TCs, climate models with anthropogenic forcing produce either no poleward migration or a much slower migration than observed (2,4,5)”.

I understood that these are important motivations of this study. However, these sentences are misinformed and should be corrected. In fact, there are more references supporting the poleward migration in the WNP in global warming scenarios (Yamaguchi et al., 2020; Kossin et al., 2016; Murakami, et al., 2012; Shen et al., 2018; Chu *et al.*, 2020). In particular, the authors cited Kossin et al (2016) to support the authors' claims, but this paper clearly demonstrates the poleward shift in the WNP in warming climate.

Yamaguchi, et al., 2020: Global warming changes tropical cyclone translation speed, *Nature Comm.*, 11(47), 1-7.

Murakami, et al., 2012: Future changes in tropical cyclone activity projected by the new high-resolution MRI-AGCM. *J. Climate*, 25, 3237–3260, doi:10.1175/JCLI-D-11-00415.1.

Chu *et al.*, 2020: Reduced tropical cyclone densities and ocean effects due to anthropogenic greenhouse warming, *Sci. Adv.*, 6, 1-10.

Shen et al., 2018: Sensitivity Experiments on the Poleward Shift of Tropical Cyclones over the Western North Pacific under Warming Ocean Conditions. *J. Meteor. Res.* 32, 560– 570

Kossin, et al., 2016: Past and projected changes in western North Pacific tropical cyclone exposure. *J. Climate.*, 29 (16), 5725-5739.

2) This paper uses ERA-I reanalysis data for TC analysis and the authors emphasize that this is the strength or originality of the present study (see below)

“(line 41-44) Moreover, uncertainty in identifying cyclogenesis positions may be larger, due to uncertain intensity estimates for developing storms (8). Confirming observed poleward trends requires other sources of track data, in which TCs are consistently identified through the record with an approach independent of those used in Best Track.”

While it is encouraging to use a new dataset independent of best track data, but it is wrong if the authors used ERA-I because of the uncertainty of best track data. In fact, best track data provide the best estimates of TC positions, intensity, and frequency using all observations, technological/analytical protocols available during a given period (Knaff et al., 2010). It is true that best track data suffer from temporal heterogeneity because of changes in observational and analytical technology such as the introduction of aircraft reconnaissance, satellites, and the Dvorak technique, but this is mostly not for the period (1979-2018) that the authors focused on. In fact, the period belongs after the geostationary-satellite and Dvorak-technique era, during which the uncertainty and heterogeneity of best track data have significantly reduced. Actually, I am rather concerned about the inaccuracies of ERA-I. Fig. 1 (left) compares the maximum wind speed (MWS) estimated from ERA-I (y axis) and best track (x axis) during the passage of all typhoons in the WNP from 2010 to 2018. Fig. 1 (right) compares best track (black) and ERA-I (gray) for the track of Typhoon Sanba in 2012. These figures show that TC intensity and track estimated from ERA-I are very inaccurate, which demonstrate that the results based on ERA-I are much less reliable compared to best track. If the authors wish to reduce the uncertainty in TC data, I recommend that the authors include additional results using different best track data (i.e., the widely used JTWC best track).

Fig. 1, Comparisons of MWS (left) and track (right) between ERA-I and best track data

Knaff JA, Brown DP, Courtney J, et al. An evaluation of Dvorak technique-based tropical cyclone intensity estimates. *Wea Forecasting* 2010;25;1362-79.

3) The authors concluded that the OND TC frequency decline is attributed to an increase in vertical wind shear in the central tropical Pacific. This conclusion is too simple and rudimentary. The OND TC frequency decline has been reported by Hsu et al. (2014). They also suggested that the recent change to a La Nina-like state induces dynamic conditions unfavorable for typhoon genesis over the southeastern WNP. Zhao et al. (2018) also reported that the number of TCs that mostly formed over the southeastern WNP (this is the cluster C1 in their study to which many OND TCs belong) was significantly modulated by the interdecadal Pacific oscillation (IPO), whose recent negative phase since 1998 corresponded to a La Niña-like sea surface temperature anomaly (SSTA) pattern. They explained that the pattern strengthened the Walker circulation in the tropical Pacific and weakened the WNP monsoon trough, suppressing genesis of C1 TCs in the southeastern WNP and predominantly contributing to the decrease in TC genesis frequency over the entire WNP basin. On the other hands, Sharmila and Walsh (2018) reported that observed poleward shift is associated with tropical expansion.

I recommend that the authors do additional analysis to investigate whether the OND TC frequency decline is related to known natural climate variabilities or anthropogenic changes such as tropical expansion. This will provide important clues for this interesting question, even if this further analysis cannot reach definitive conclusions.

Hsu et al. (2014) An Abrupt Decrease in the Late-Season Typhoon Activity over the Western North Pacific, *J. Climate* 27(11), 4296–4312

Zhao et al., 2018: Contribution of the Interdecadal Pacific Oscillation to the Recent Abrupt Decrease in Tropical Cyclone Genesis Frequency over the Western North Pacific since 1998, *J. Climate* 31, 8211–8224

Sharmila, S., & Walsh, K. J. E., 2018: Recent poleward shift of tropical cyclone formation linked to Hadley cell expansion. *Nature Climate Change*, 8(8), 730-736.

4) The authors have not introduced important previous studies related to this paper. Hsu et al. (2014) reported the OND TC frequency decline in the western North Pacific for the first time, but this has not been introduced. As another example, this paper argued that the observed poleward shift of TCs is primarily related to changes in TC frequency, by decomposing the annual-average statistics into seasonal components. However, the role of TC frequency in the poleward shift (including decomposition method) was first introduced by Moon et al. (2015), but this was also omitted in the reference. This is important because readers believe that this paper is the first to report the result and used the method for the first time, unless the relevant previous findings and methods are introduced.

Hsu et al. (2014) An Abrupt Decrease in the Late-Season Typhoon Activity over the Western North Pacific, *J. Climate* 27(11), 4296–4312

Moon, et al. (2015). Roles of interbasin frequency changes in the poleward shifts of the maximum intensity location of tropical cyclones, *Environ. Res. Lett.*, 10 (2015) 104004.

Other concerns

1) The authors defined October-December as autumn, but in the western North Pacific, December is not autumn. Hsu et al. (2014) defined TCs during OND as late-season TCs. The authors need to think about more proper terms about OND

Author's Response to Reviewer:

Reviewer #1 (Remarks to the Author):

Review of "Poleward migration of western North Pacific tropical cyclones related to changes in cyclone seasonality" for Nature Communications manuscript NCOMMS-20-46326-T

Recommendation

The authors analyze the poleward shift of typhoons using Best Track data from Japan with further support from objectively tracked reanalysis cyclone tracks. While this trend has garnered attention lately as a signal for a connection to climate change via an expanding Hadley cell circulation. The authors decompose the trend into two seasonal portions – summer (JAS) and autumn (OND) – showing that while the summer portion has undergone some change, the autumn portion has experienced a significant decrease in relative numbers leading to the noted shift in latitude. The details are interesting and tell a bit more complex exists in the poleward shift. Given the recent interest in this topic, the rigor of the analysis and the results, I would recommend acceptance with minor revision.

-- We appreciate that the reviewer recognizes the research value of our paper. The reviewer's comments have been carefully considered and addressed as follows.

Specific comments

1. Title – Since tropical cyclones in this region are only called typhoons, the title could be more succinctly as "Poleward migration of typhoons related to changes in cyclone seasonality"

-- We thank the reviewer for pointing this out. We agree that in the Northern West Pacific tropical cyclones are often called 'typhoons'. But, by a strict definition, only those storms reaching typhoon intensify, i.e., maximum sustained wind speed ≥ 64 knots, are classified as typhoons. In our paper, we focus on tropical cyclones reaching severe tropical storm intensity or higher (maximum sustained wind speed ≥ 48 knots). We think by definition, at least, it would be more precise to call these storms tropical cyclones rather than typhoons. We have added clarification to address this comment. Please see Lines 75-77 in the main text and Lines 473-475 in the Methods section

2. Abstract – is clear and well written

-- We thank the reviewer for this. We made some minor changes in the abstract in the revised version to reflect the improvement of the manuscript.

3. Line 74 – perhaps cyclolysis vice lysis?

-- We thank the reviewer for this valuable comment. We defined 'TC lysis' as the short name of tropical cyclolysis in Line 91.

4. Line 95 – the series of numbers are less clear in this sentence ... might be easier to read if written as "... LMI (42+-31), and cyclogenesis (38+-31) migrated ... all track points (70+-35) .." [where I write +- for plus or minus ... not sure where that is on my keyboard).

-- We are sorry for the confusion. We have rewritten this sentence. Please see lines 114-117 in the revised version.

5. Line 106 – The statement that ERA Interim trends are larger is confusing because I first looked at the lower portion of Table 1. ... I think it would be better to separate Table 1 into 2 tables. One for each of the different periods of record, for clarity.

-- We are sorry for the confusion and thank the reviewer for this useful suggestion. We have separated Table 1 into two tables (one for Best Track and one for ERA-Interim). The Best Track table has two subgroups: one for the satellite era and one for the pre-stellate era; the ERA-Interim table is for the satellite era only. Both tables have headers explicitly giving the distinction. We hope this change will address the reviewer's concern.

6. Paragraph at line 146 – I must object to this use of parentheses to denote opposites. As described in AGU Eos, they do not add to readability, in fact, more the opposite. Please rewrite this paragraph. Specifically, they state: "Making a journal article a few words longer for clarity and to avoid confusion is well worth the expense in extra bits of storage." Why? I am forced to read the paragraph twice and mentally note that the parentheses at lines 146, 151, 156 and 160 (second set) are examples or figures while the parentheses at lines 147, 149, 151 (2nd group), 152, 153, and 160 (1st set) are opposites. Such intermingling would cause syntax errors were this programming ... and to that extent, readers are not programs that can mentally carry the argument of all the opposites along. Another sentence or two would make your points clearer than the seemingly efficient usage of parentheses.

<https://eos.org/opinions/parentheses-are-not-for-references-and-clarification-saving-space>

-- We agree with the reviewer that using parentheses to mean the opposite is confusing sometimes, especially when parentheses are also being used for clarification in the same paragraph/paper. We appreciate that the reviewer highlights the problem. We have removed all parentheses used for meaning opposites throughout the manuscript, and instead added sentences to make our points clearer (for example, see Lines 181-186 in the revised version).

7. Line 171 ... same line has multiple uses for parentheses. For instance, the sentence could be written "For a given year, dP_{Lat} is proportional to the correlation of p' and lat' (that is, positive for same-signed variations in p and lat)." .. Assuming "same signed variations" is akin to correlation.

-- We thank the reviewer, again, for the suggestion to improve the readability by better using parentheses. We have rewritten this sentence. Please see lines 206-208 in the revised version.

8. For overall readability, please consider replacing "JAS" with "summer" and "OND" with "autumn" ... these phrases are used in introducing the topic on lines 120 and 122 ... but the author seems to prefer JAS and OND. While the acronym is specific, the words are more readable and clearly defined in lines 120/122 so why not use them.

-- This is a valuable comment. In the revised version, we redefined the seasons of JAS and OND as 'peak season' and 'late season' for West Pacific TCs (Lines 150-152), respectively. Then, following the reviewer's comment, we used 'peak season' and 'late season' for JAS and OND throughout the paper where applicable. We hope this change will improve the paper's readability.

9. Line 415 – should be "Values in parentheses..."

-- Thanks! We have modified the table captions after separating the original Table 1 into two tables. The parentheses were removed in the revised version.

10. Figure 1a – would be clearer if the left hand y axis only extended down to 2000km (since the red & blue lines correspond to the proportion on the right hand axis). Likewise, the proportion

axis could stop at 80%. Other figures in the (and in supplement) could likewise be adjusted for clarity. [Fig 5c has a nice demonstration of this.]

-- We thank the reviewer for another useful comment. We have adjusted the y axis in all figures where possible.

11. Figure 5a – Put the MDR in this plot as well.

-- Thanks for pointing this out. The MDR box was added in all subplots of Figure 5 and Extended Data Figure 9.

Reviewer #2 (Remarks to the Author):

Review of “poleward migration of western North Pacific tropical cyclones related to changes in cyclone seasonality”

GENERAL COMMENTS

1. This is an interesting paper to extend previous studies on poleward migration of tropical cyclones for different seasons. While the idea is good and innovative, it is unclear as to how the results are obtained because of the following:

- a. lack of the definition of the metrics (Specific Comments 1, 9 and 10), and
- b. the use of the ERA-Interim dataset as the primary source of data when this dataset is well known for many uncertainties – see Specific Comment 4.

-- We appreciate that the reviewer recognizes the research value of our paper. We have carefully considered and addressed these two main comments in the revised version of the manuscript. Please see our detailed response below. In short, a) we added new sentences in the Main text and a subsection in the Methods, to explicitly define the TC position metrics (please see Lines 87-94 in the main text and Lines 551-572 in the Methods section, in revised version); b) we included additional sources of Best Track (JTWC and CMA data) in our analysis; c) we added a cautious statement on ERA-Interim in the main text (Lines 49-52) and the Methods section (Lines 534-549) with advantages and caveats clearly mentioned.

2. With the problems associated with the ERA-I dataset, why is the best-track dataset from JMA or JTWC not used as the primary source of data for the study?

-- We thank the reviewer for pointing this out. There might be some misunderstanding of the use of ERA-Interim in our analysis, probably due to insufficient clarity in the first version of the manuscript. The Best Track data are the main source of data used to derive our conclusions; ERA-Interim TC data are used as an independent data source to support the results from Best Track. We have added a sentence in Lines 49-52 to clarify this.

-- In the revised version, we include two additional sources of Best Track data (JTWC and CMA). These data were used throughout the analysis. For example, in the revised version, in Figure 6, Extended Data Figures 10-11, and related text, we used both Best Track data and ERA-Interim data, to reveal the possible causes of the reduction of late-season (OND) TC frequency. We think the above changes will address the reviewer’s comment on ERA-Interim TC data.

3. Throughout the manuscript, no result of statistical test of any of the trends was provided except in Table 1. It is therefore not clear which of the trends are statistically significant.

-- We thank the reviewer for pointing this out. In the previous version of our paper, the trend

values came with error bars (e.g., 87 ± 40 km/decade); the error bars represent the 95% confidence interval. We meant to (but obviously did not) state this in the first version of the manuscript. In the reversed version, we have added a ‘Statistical analyses’ subsection in Methods section, to describe the statistical tests used for both the trend and correlation. Please see Lines 594-610.

4. Normally, these deficiencies, together with those mentioned below in the Specific Comments, would lead to rejection of the manuscript. However, given the innovative idea on a possible seasonality of poleward migration, and hence how such seasonality might contribute to the overall migration trend, if any, it is recommended that the authors address all these issues mentioned in this review, re-do the analyses using the JTWC or JMA dataset, compare the results in the pre- vs post-satellite eras, and then re-submit the manuscript.

-- We have carefully considered these comments. We included three sources of Best Track data (JTWC, JMA and CMA; see Lines 100-102 in the main text and Lines 467-494 in the Methods section) in our analysis, and we also adjusted the statement on the advantages and caveats of ERA-Interim TC data (please see Lines 49-52 and Lines 534-549).

-- In the revised version, the analysis for the pre-satellite era is cautiously compared with that in the latest decades, as the reviewer suggested. The results were rewritten in Lines 235-270. We hope these improvements will address the reviewer’s comments.

SPECIFIC COMMENTS

1. Lines 73-74:

(a) Throughout the manuscript, there is no definition of cyclogenesis, LMI and lysis. For example, is genesis defined as the first point of the TC reaching 25 or 30 kt? For LMI, in many situations, the LMI can be maintained for a period of time, and in some cases, a TC can weaken briefly after reaching the LMI and then re-intensify to the LMI again. So which position is defined as the latitude of LMI? Similarly, is the lysis position the last position before dissipation irrespective of intensity? What about cases of extratropical transition? Many of the TCs dissipate over land but not over the ocean. Are land points considered? All these definitions and how the data points are obtained must be clearly presented and explained so that the reader can compare the results with those from previous studies.

-- We thank the reviewer for these comments. In the revised version, we have added a brief description on these metrics in the main text (Lines 87-94) and further added a subsection *TC location metrics* in the Methods section (Lines 551-572) to carefully define these metrics.

-- In short, the latitude of LMI is determined by the first point where each TC reaches its lifetime-maximum intensity. In Best Track, the latitude of cyclogenesis is determined by the first track point whose intensity reaches 34 knots, while the cyclolysis location is the last track point in observations. In ERA-Interim, cyclogenesis and cyclolysis are the first and last point, respectively, of the identified TC track where the relative vorticity exceeds the given thresholds (see Methods for detail). We also added a paragraph in the Methods to suggest the uncertainties in defining these metrics when using Best Track (please see Lines 477-484).

(b) The net latitudinal transit distances need to be clearly explained. Recurving TCs have shorter transit distances, and hence the net latitudinal displacement could be less.

-- The reviewer is right. Different types of TCs (e.g., recurving TCs and straight-moving TCs) may have different transit distances, which may in turn affect the trend detection. On seasonal timescales, the net latitudinal transit distances are larger in summer and shorter in winter. In this manuscript, we are trying to make a point that the observed changes in the net latitudinal transit displacement are mostly caused by the changes in TC seasonality.

-- We thank the reviewer for pointing this out. We have added a subsection (*TC location metrics* in the Methods) to clarify the net latitudinal transit distances. Please see Lines 563-572.

2. Line 78: Please explain how to have >800 samples per year.

-- We thank the reviewer for mentioning this point. We have modified this sentence ('>500 track points for ~20 TCs per year, on average') to address this comment. Please see Line 97. Because TC lifetime is shorter in the revised version (by restricting the first point of storm as the point reaching 34 knots; see Lines 120 and 555-557), the sample size reduces from 800 to 500.

3. Line 81: Data from 1951 to the pre-geostationary satellite era (approximately 1975) have been criticized as having many problems, especially for intensity, which implies large uncertainty in the determination of the position of LMI. This problem therefore has to be discussed as to how it is taken into consideration in the subsequent analyses and evaluation. Data in the post-satellite era should be studied independently to see if the conclusions are different.

-- We thank the reviewer for this useful comment. We agree that Best Track data in the pre-geostationary satellite era have large uncertainty and errors in TC intensity and position estimates. The spatial and temporal inhomogeneity in observed tracks could lead to an abrupt change in TC track locations, both for absolute locations (e.g., genesis) and relative locations (e.g., transit distance).

-- As the reviewer suggested, in the revised version, we estimated the trends separately for the satellite (1979-2018) and pre-satellite eras (1951-1978). These are reflected in Table 1 and Extended Data Table I, and Figure 4 and Extended Data Figures 7&8. We have added discussion of the estimated trends for the pre-satellite era, including the possible errors affecting the trend detection. We also recall the caution about the reliability of the trends for early years. Please see Lines 235-270.

4. Lines 81-82: Estimating the metrics mentioned in lines 72-73 using ERA-Interim has a number of problems as well. This dataset does not have the horizontal resolution to give a good estimate of intensity, and hence the latitude of LMI determined from the dataset is likely incorrect. Figure 2c and Extended Data Figure 3c show consistent differences between the best track data and the ERA-I dataset, which suggests that the latter has some inherent problem in providing a good estimate of LMI and its latitude. Further, this dataset tends to give more vortices than the observed, especially near the dateline. This can actually be seen in Fig. 2a, Extended Data Figure 3a, and the location of the black dots in Fig. 5 of the manuscript. Thus, while ERA-I could be used to provide some estimate of the metrics, discussions of the results from this dataset must include caveats on these uncertainties. In fact, it is not clear why the authors choose this dataset over the best-track dataset.

-- We thank the reviewer for this comment. We did not mean to use ERA-Interim TC track in preference to the Best Track. Instead, we used ERA-Interim data as an independent data source to support the conclusions made from the Best Track. To avoid the confusion, in the revised version, we made a statement in the main text (Lines 49-52).

-- In the revised version, in understanding the relationships between large-scale environments and the frequency of WNP TC genesis, we used both Best Track and ERA-Interim TC data (Figure 6 and Extended Data Figure 11, Lines 278-283). Environmental factors are correlated with TC genesis either from Best Track or ERA-Interim (also Lines 307-310).

-- We also added a paragraph in the Methods section to explicitly highlight the merits and caveats of ERA-Interim TC track in trend detection, including the displacement of LMI positions. Please see Lines 534-549.

5. Lines 82-83: What is meant by “an extended lifecycle”? Why is the best-track dataset not able to provide the same information?

-- We thank the reviewer for the comments. Our objective TC tracking algorithm can produce the most coherent tracks and retains the full lifecycles of the systems, including their pre-TC and post-TC stages. These extended stages are defined by TC dynamical structures (e.g. warm core and vorticity) rather than by TC intensity as in Best Track. ERA-Interim data can track the genesis position further backward, and track the lysis location further forward (please see ‘ERA-Interim TC track data’ in the Methods section).

-- Best Track intensity data are uncertain, especially for weak TCs and weak stages of the TC lifecycle. For example, if we choose the first point of the Best Track track as TC genesis, different Best Track data sources give us very different trends for genesis. To reduce this uncertainty in trend detection in Best Track, similar to Daloz and Camargo (2018), we only select severe tropical storms (≥ 48 knots) and the stages of the storm lifecycle since reaching tropical storm intensity (≥ 34 knots). This means that Best Track data have a shorter lifecycle for storms, relative to the ERA-Interim dataset. We added sentences to clarify these points in the revised version of the manuscript where applicable. Please see Lines 75-77, 137-140, 473-475 and 555-557.

6. Lines 86-87: Please explain how ENSO and PDO are actually removed using multiple linear regression for each of the metrics examined.

-- Sorry for the confusion. We added a paragraph in the Methods to explain how we did this. Please see Lines 605-610.

7. Lines 88-89: It is doubtful that the conclusions can be “identical”. No difference at all?

-- In this revised version, we explicitly showed the differences in trends between three Best Track sources, by adding sentences (Lines 127-133, and 227-231) and an additional table (Extended Data Table 1). We think these improvements will address this comment.

8. Line 97: What is meant by “due to small trends in the 2010s”?

-- Sorry for the confusion. We removed this sentence.

9. Line 135: “TC lifetime”? A new metric? What is the definition?

-- Thanks for pointing this out. We meant to say ‘TC duration’. In the revised version, we modified this sentence to clarify it (see Line 167).

10. Lines 381-383: What is meant by “monthly latitude”? Please also give an equation for the definition of $p(m, i)$. In fact, it is not clear how Equation (1) can be rewritten to Equation (2). Please explain clearly why this can be done.

-- We clarified the definitions of monthly latitude and monthly proportion in the revised version, and added an additional equation explaining the transformation of the original Eq(1) to Eq(2). Please see the changes in ‘Decomposing the annual-mean latitude of TC location’ in the Methods section.

11. Table 1: Why are there no entries for LMI for the best-track dataset?

-- In the early period (1951-1978) of JMA best track, the latitudinal trend of LMI is not calculated, because TC intensity estimates are not recorded in the JMA data (which only provide TC category values). We added sentences in both the Methods (Lines 489-494 and 569-572), and captions of Table 1 and Extended Data Table 1, to clarify this.

Reviewer #3 (Remarks to the Author):

This paper claimed that the shift of average location of observed western North Pacific (WNP) tropical cyclones (TCs) over the last four decades is associated with relatively more (less) frequent TC genesis in summer (autumn). The authors also concluded that the TC frequency decline in autumn is attributed to the increase in vertical wind shear in the central tropical Pacific. This paper has a merit in that it offers a different perspective on long-term trends in TC activity, by decomposing the annual-average statistics into seasonal components (as authors mentioned). However, unfortunately there are some issues that need to be resolved before this manuscript can be acceptable for publication, as summarized below.

-- We thank the reviewer for the valuable comments and suggestions. The reviewer's comments have been carefully considered and addressed as follows.

Major concerns

1) The authors claimed that "(line 7-8) While anthropogenic forcing is hypothesized to be the cause, climate-model simulations do not reproduce this trend in response to anthropogenic forcing" and "(line 32-34) However, for WNP TCs, climate models with anthropogenic forcing produce either no poleward migration or a much slower migration than observed (2,4,5)".

I understood that these are important motivations of this study. However, these sentences are misinformed and should be corrected. In fact, there are more references supporting the poleward migration in the WNP in global warming scenarios (Yamaguchi et al., 2020; Kossin et al., 2016; Murakami, et al., 2012; Shen et al., 2018; Chu et al., 2020). In particular, the authors cited Kossin et al (2016) to support the authors' claims, but this paper clearly demonstrates the poleward shift in the WNP in warming climate.

Ref;

Chu et al., 2020: Reduced tropical cyclone densities and ocean effects due to anthropogenic greenhouse warming, Sci. Adv., 6, 1-10.

Kossin, et al., 2016: Past and projected changes in western North Pacific tropical cyclone exposure. J. Climate., 29 (16), 5725-5739.

Murakami, et al., 2012: Future changes in tropical cyclone activity projected by the new high-resolution MRI-AGCM. J. Climate, 25, 3237–3260, doi:10.1175/JCLI-D-11-00415.1.

Nakamura, J. et al. 2017; Western North Pacific tropical cyclone model tracks in present and future climates. J. Geophys. Res.122, 9721–9744.

Shen et al., 2018: Sensitivity Experiments on the Poleward Shift of Tropical Cyclones over the Western North Pacific under Warming Ocean Conditions. J. Meteor. Res. 32, 560–570

Yamaguchi, et al., 2020: Global warming changes tropical cyclone translation speed, Nature Comm., 11(47), 1-7.

--Thank the reviewer for pointing this out. We agree with the reviewer that literature indicates that anthropogenic forcing could lead to a poleward shift of TC tracks. Here, we are not ruling out this possibility. Instead, we argue that tropical expansion due to a warming climate may not fully explain the observed rate of poleward shift of WNP TCs, and that other factors need to be considered. In the revised version, we modified the description on this argument. Please see Lines 31-36.

-- We thank the reviewer for suggesting these relevant references. We carefully went through these references, and we cited the most relevant ones in the revised version of the manuscript. We found that most of these references support our argument that 'although some GCMs can qualitatively produce a poleward shift of TC track (LMI) under anthropogenic forcing, but

none of these GCM models can quantitatively reproduce the rate of poleward shift in observations.’ Some key points from these references are:

- Kossin et al. (2016) shows a poleward shift (about 20km/decade) of LMI of WNP TCs in GCM simulations under anthropogenic forcing. Shen et al (2018) also shows that in an idealized experiment of WRF, SST warming can cause a northward shift of WNP TCs. Given the SST warming trend of 0.13 deg.C/decade (Wu et al. 2020), the poleward trend in TC LMI in Shen et al (2018) is estimated about 3-20 km/decade. As we have argued above, these trend values in GCM models are much smaller than the observed LMI trend, which is about 60-100 km/decade over the modern observational era.
- Yamaguchi et al. (2020) mentioned “Nakamura et al. (2017) showed through multi-model analyses that there is a large range of uncertainty across numerical simulations as to whether a poleward shift of TC occurrence exists in the western North Pacific basin, where there has been the largest trend in the observations for the lifetime maximum intensity of TCs and TC genesis location”. Both papers agree there is a large uncertainty in projected TC track shifts in a warming climate. Nakamura et al. (2017) mentioned that the variance of TC latitudinal distance relative to the track centroid in WNP under a warming climate is increased significantly in a multi-model analysis. However, for the absolute latitudes of TC tracks, such as the track mean location (centroid) or LMI location, there are no robust northward shifts.
- Murakami et al. (2012) found a northward shift of TC track in a restricted band of Pacific (180W-120W), but the trend is not evident in the WNP, in either 20 or 60KM GCM simulations forced with climate warming. Similarly, Chu et al (2020) find no direct evidence that a poleward shift of LMI or genesis is caused by anthropogenic warming. Thus, although some of these GCM runs qualitatively support a poleward shift of TC track (LMI), none of these GCMs quantitatively reproduce the observed rate of poleward shift.
- It is interesting to see that Yamaguchi et al. (2020) and Chu et al (2020) both highlight the relative decrease of TC frequency in the tropics and a relative increase in the extra-tropics in a warmer climate. This may indicate that the relative frequency of summer TCs has increased, as summer TCs normally form and travel further northward. However, unfortunately, these authors did not evaluate changes in TC seasonality.

2) This paper uses ERA-I reanalysis data for TC analysis and the authors emphasize that this is the strength or originality of the present study (see below)

“(line 41-44) Moreover, uncertainty in identifying cyclogenesis positions may be larger, due to uncertain intensity estimates for developing storms (s). Confirming observed poleward trends requires other sources of track data, in which TCs are consistently identified through the record with an approach independent of those used in Best Track.”

While it is encouraging to use a new dataset independent of best track data, but it is wrong if the authors used ERA-I because of the uncertainty of best track data. In fact, best track data provide the best estimates of TC positions, intensity, and frequency using all observations, technological/analytical protocols available during a given period (Knaff et al., 2010). It is true that best track data suffer from temporal heterogeneity because of changes in observational and analytical technology such as the introduction of aircraft reconnaissance, satellites, and the Dvorak technique, but this is mostly not for the period (1979-2018) that the authors focused on. In fact, the period belongs after the geostationary-satellite and Dvorak technique era, during which the uncertainty and heterogeneity of best track data have

significantly reduced. Actually, I am rather concerned about the inaccuracies of ERA-I. Fig. 1 (left) compares the maximum wind speed (MWS) estimated from ERA-I (y axis) and best track (x axis) during the passage of all typhoons in the WNP from 2010 to 2018. Fig. 1 (right) compares best track (black) and ERA-I (gray) for the track of Typhoon Sanba in 2012. These figures show that TC intensity and track estimated from ERA-I are very inaccurate, which demonstrate that the results based on ERA-I are much less reliable compared to best track. If the authors wish to reduce the uncertainty in TC data, I recommend that the authors include additional results using different best track data (i.e., the widely used JTWC best track).

Ref: Knaff JA, Brown DP, Courtney J, et al. An evaluation of Dvorak technique-based tropical cyclone intensity estimates. Wea Forecasting 2010;25;1362-79.

Fig. 1, Comparisons of MWS (left) and track (right) between ERA-I and best track data

-- We appreciate that the reviewer thinks carefully and constructively about the use of the ERA-Interim TC dataset. We agree that ERAI has problems representing the absolute values of TC intensity and location. But the accuracy of trends and variability (on both seasonal and interannual timescales) of TC location metrics is not necessarily determined by the accuracy of the absolute values of these metrics. Actually, the trends and variability are quite well reproduced in ERA-Interim, compared to Best Track. We use ERA-Interim TC data as an independent data source to increase our confidence in the trends detected from Best Track, rather than choosing ERA-Interim over Best Track. In the revised version of the manuscript, we made a statement in the main text (Lines 49-52). We also added a paragraph in the Methods to explicitly highlight the merits and caveats of ERA-Interim TC track in trend detections (Lines 534-549).

-- We also thank the reviewer for suggesting including more sources of Best Track data, to reduce e.g., the interagency uncertainty in trend detection. In the new version of the manuscript, we use three sources (JTWC, JMA and CMA) of Best Track data. We create a Best Track ensemble for a variety of TC location metrics to evaluate the robustness of the poleward shift of WNP TCs. Please see Lines 100-102, and 'Best Track data' subsection in the Methods for details (Lines 467-494). All the figures and tables are updated with the ensemble results. We also added a few sentences to shed light on the uncertainties which can potentially affect the detection of TC trends, for which please see Lines 127-133 in the main text and Lines 477-484 in the Methods section.

3) The authors concluded that the OND TC frequency decline is attributed to an increase in vertical wind shear in the central tropical Pacific. This conclusion is too simple and rudimentary. The OND TC frequency decline has been reported by Hsu et al. (2014). They also suggested that the recent change to a La Nina-like state induces dynamic conditions unfavorable for typhoon genesis over the southeastern WNP. Zhao et al. (2018) also reported that the number of TCs that mostly formed over the southeastern WNP (this is the cluster C1 in their study to which many OND TCs belong) was significantly modulated by the

interdecadal Pacific oscillation (IPO), whose recent negative phase since 1998 corresponded to a La Niña-like sea surface temperature anomaly (SSTA) pattern. They explained that the pattern strengthened the Walker circulation in the tropical Pacific and weakened the WNP monsoon trough, suppressing genesis of C1 TCs in the southeastern WNP and predominantly contributing to the decrease in TC genesis frequency over the entire WNP basin. On the other hands, Sharmila and Walsh (2018) reported that observed poleward shift is associated with tropical expansion.

I recommend that the authors do additional analysis to investigate whether the OND TC frequency decline is related to known natural climate variabilities or anthropogenic changes such as tropical expansion. This will provide important clues for this interesting question, even if this further analysis cannot reach definitive conclusions.

Ref:

Hsu et al. (2014) *An Abrupt Decrease in the Late-Season Typhoon Activity over the Western North Pacific*, *J. Climate* 27(11), 4296–4312

Zhao et al., 2018: *Contribution of the Interdecadal Pacific Oscillation to the Recent Abrupt Decrease in Tropical Cyclone Genesis Frequency over the Western North Pacific since 1998*, *J. Climate* 31, 8211–8224

Sharmila, S., & Walsh, K. J. E., 2018: *Recent poleward shift of tropical cyclone formation linked to Hadley cell expansion*. *Nature Climate Change*, 8(8), 730-736.

-- We thank the reviewer for encouraging us to deepen the analysis of TC seasonality change, e.g., the decline of OND TC frequency. In the revised version of the manuscript, we looked at TC-related large-scale environments more broadly, including SST, lower-level vorticity, winds at upper and lower levels, mid-level humidity, etc. The main conclusions are: 1) we confirmed findings in previous studies that the changes in these large-scale environments are associated with the strengthening of the Pacific Walker Circulation and the increased west-to-east SST gradient; 2) we further emphasised that these changes have a seasonally non-uniform effect on the WNP TC occurrence, i.e., significantly suppressing late-season (OND) TCs and barely changing peak-season (JAS) TCs; and 3) however, the exact roles played by other longer-term climate variability (e.g. IPO and AMO; here, the ENSO and PDO impacts are eliminated) and anthropogenic forcing in driving such changes remain not fully understood.

-- Please see Lines 283-338, Figures 5-6, and Extended Data Figures 9&11, for the updated analysis and results. We think this additional analysis will address the reviewer's comment and significantly improve the quality of the paper.

4) The authors have not introduced important previous studies related to this paper. Hsu et al. (2014) reported the OND TC frequency decline in the western North Pacific for the first time, but this has not been introduced. As another example, this paper argued that the observed poleward shift of TCs is primarily related to changes in TC frequency, by decomposing the annual-average statistics into seasonal components. However, the role of TC frequency in the poleward shift (including decomposition method) was first introduced by Moon et al. (2015), but this was also omitted in the reference. This is important because readers believe that this paper is the first to report the result and used the method for the first time, unless the relevant previous findings and methods are introduced.

Ref:

Hsu et al. (2014) *An Abrupt Decrease in the Late-Season Typhoon Activity over the Western North Pacific*, *J. Climate* 27(11), 4296–4312

Moon, et al. (2015). *Roles of interbasin frequency changes in the poleward shifts of the maximum intensity location of tropical cyclones*, *Environ. Res. Lett*, 10 (2015) 104004.

-- We thank the reviewer for suggesting these useful references. We have cited these references properly in our revised version, for which please Lines 208-210 and 286.

Other concerns

1) The authors defined October-December as autumn, but in the western North Pacific, December is not autumn. Hsu et al. (2014) defined TCs during OND as late-season TCs. The authors need to think about more proper terms about OND TCs.

-- We thank the reviewer for pointing this out. In our revised version, following Hsu et al. (2014), we defined OND as the late season of WNP TCs, and JAS as the peak season (please see Lines 150-152). These terms have been used throughout the manuscript.

REVIEWER COMMENTS

Reviewer #1 (Remarks to the Author):

The authors have sufficiently responded to my comments. I believe the paper is improved.

Reviewer #2 (Remarks to the Author):

Second review of "Poleward migration of western North Pacific tropical cyclones related to changes in cyclone seasonality"

GENERAL COMMENTS

1. The authors did a very good job in revising the manuscript to address most of my comments. However, there are still a number of points that need to be considered, which will be detailed below.
2. On the physical mechanism part, a number of studies have already proposed explanations for the decrease in late-season TC activity. These results do not give any new insights.
3. Recommendation: accept with minor revisions

SPECIFIC COMMENTS

1. Lines 10-12: This sentence needs to be rewritten. The fact that late-season TCs become less frequent does not "cause" higher-latitude TCs to contribute more. Because late-season TCs are mostly at low latitudes, a decrease in its frequency of occurrence leads to a lower contribution to the annual average over the entire ocean basin.
2. Line 13: What is meant by "inconsistent" effect? Inconsistent compared with what?
3. Line 89: In the Methods section, TCs with maximum winds less than 34 kts are not counted. Does "genesis" mean the first track point with 34 kts in maximum winds?
4. Line 91: The "last track point": how is extratropical transition handled? Does the last track point include all points with extratropical transition?
5. Lines 91-92: I asked about straight-moving vs recurving TCs in calculating the net transit distance. I don't think the authors answered this well in their response. Please explain how this is handled or considered.
6. Lines 107-8: In removing the ENSO and PDO impacts, how is the multiple regression done? For monthly results, is the multiple regression carried out using the ENSO and PDO indices for the same month? And for annual averages, are the annual indices used? Also, are ALL the data presented in this paper treated this way?
7. Lines 153-160: The terms "significant" and "statistically significant" have been used many times. Decreasing or increasing trends have also been mentioned. Please provide the levels of significance in all these discussions.
8. Lines 310-12: The suggestion that the unfavourable atmospheric conditions in the MDR are related to a strengthening of the Walker Circulation is questionable. The longitude-height cross-section in Figure 5d is for the average over 5S-5N, which is not over the MDR. How are these two related?

Reviewer #3 (Remarks to the Author):

[They have had no further comments]

Author's Responses to Reviewers

Reviewer #2 (Remarks to the Author):

Second review of "Poleward migration of western North Pacific tropical cyclones related to changes in cyclone seasonality"

GENERAL COMMENTS

1. The authors did a very good job in revising the manuscript to address most of my comments. However, there are still a number of points that need to be considered, which will be detailed below.

-- We appreciate that the reviewer recognizes the improvement of our revised manuscript and for the additional useful comments. The reviewer's comments have been carefully considered and addressed as follows.

2. On the physical mechanism part, a number of studies have already proposed explanations for the decrease in late-season TC activity. These results do not give any new insights.

-- We thank the reviewer for this comment. We agree that the causes of decline of late-season TC frequency were reported previously. As mentioned in our manuscript (Lines 303-306, including the references), this change is likely related to the strengthening of the Pacific Walker Circulation associated with an enhanced west-to-east SST gradient. However, our analysis further highlights the different effects of the strengthening of the Pacific Walker Circulation on TC activity for different seasons and not just the late season. In other words, the changes associated with the Pacific Walker Circulation affect more the late-season WNP TCs than the peak-season TCs, which is the key to the seasonal redistribution of TC frequency and consequently to the poleward migration of annual TC location. From this viewpoint, we think our analysis provides a new hypothesis of long-term changes in TC activity. We have clarified this point in Lines 308-310.

3. Recommendation: accept with minor revisions

SPECIFIC COMMENTS

1. Lines 10-12: This sentence needs to be rewritten. The fact that late-season TCs become less frequent does not "cause" higher-latitude TCs to contribute more. Because late-season TCs are mostly at low latitudes, a decrease in its frequency of occurrence leads to a lower contribution to the annual average over the entire ocean basin.

-- We thank the reviewer for pointing this out. We modified this sentence to make this point more precise. Please see Lines 10-12 in the latest revised version of manuscript.

2. Line 13: What is meant by "inconsistent" effect? Inconsistent compared with what?

-- We thank the reviewer for this comment. We meant to say 'different effects on late- and peak-season TC frequency'. We rewrote this sentence to make it clearer. Please see Lines 12-14.

3. Line 89: In the Methods section, TCs with maximum winds less than 34 kts are not counted. Does “genesis” mean the first track point with 34 kts in maximum winds?

-- Yes, in our analysis, TC genesis is defined as the 1st track point with maximum winds ≥ 34 knots. This means that the track points with maximum winds < 34 knots are excluded here. We changed Line 89 to address this clarification. We refer the reader to the Methods section for more details (Lines 575-577).

4. Line 91: The “last track point”: how is extratropical transition handled? Does the last track point include all points with extratropical transition?

-- Thanks for pointing this out. In Best Track, the extratropical cyclone stage is included, but it is considered differently by different meteorological agencies. For example, the years when extratropical cyclone tracks begin to be included are 2004 in JTWC data, 1977 in JMA data and further earlier in CMA data. It is not straightforward to reduce the uncertainties in location metrics related to the different treatments of extratropical cyclones. Actually, this forms partially the spread of location metrics in the Best Track ensemble. We added statement in the Methods section to clarify this point. Please see Lines 503-504 and 578-580.

5. Lines 91-92: I asked about straight-moving vs recurving TCs in calculating the net transit distance. I don't think the authors answered this well in their response. Please explain how this is handled or considered.

-- We thank the reviewer for this comment. We meant to (but obviously did not) address this comment in more detail in the last version of the manuscript. We are sorry for that. In this new revision, we classified WNP TCs into recurving and straight-moving TCs, following Zhang et al., (2013, DOI: <https://doi.org/10.1175/JAMC-D-12-045.1>). This is mentioned in Lines 94-96 in this new revision, with detailed process described in Lines 596-600. The climatology of frequency and location for each type of TC tracks was provided in Extended Data Figure 7. The trends in monthly frequency (and relative frequency) of recurving and straight-moving TCs were provided in Extended Data Figure 8. We also calculated the poleward trends of location metrics, for each type of TC tracks (Extended Data Table 2).

-- We found no significant change in the proportions of either recurving TCs or straight-moving TCs, suggesting no significant recurvature changes in WNP TCs. Instead, we found that straight-moving TCs become significantly fewer in late season, contrasting with no significant changes in recurving TC frequency across all seasons. This corresponds to the larger changes in straight-moving TC seasonality ($\delta_{p'}$), which are responsible for the larger poleward trends in straight-moving TCs. This is consistent with our main conclusion that TC seasonality changes strongly affect the trend detection in annual-mean statistics. We also added a paragraph to address this comment. Please see Lines 241-253.

6. Lines 107-8: In removing the ENSO and PDO impacts, how is the multiple regression done? For monthly results, is the multiple regression carried out using the ENSO and PDO indices for the same month? And for annual averages, are the annual indices used? Also, are ALL the data presented in this paper treated this way?

-- Thanks for this comment. When removing the ENSO and PDO impacts, we used the same ENSO and PDO indices for both monthly and annual results. Yearly ENSO and PDO indices are taken as averages of the monthly indices over the typhoon season July–November when most of TCs occur. We don't think using the monthly climate indices with the same months as the analyzed data (TC or environments) will make a major change in our results/conclusions. This is because very few TCs occur outside typhoon season (July–November), and because the interannual-to-decadal variability of ENSO and PDO doesn't change substantially from month to month. We modified the sentences for this procedure in Method to clarify this point. Please see Lines 634-642.

7. Lines 153-160: The terms “significant” and “statistically significant” have been used many times. Decreasing or increasing trends have also been mentioned. Please provide the levels of significance in all these discussions.

-- Thanks for pointing this out. In the Methods section, we mentioned that in our paper the statistically significant values (either trend or correlation) are at the 95% confidence level, unless stated otherwise. To recall this point, we added a sentence just before the result section in the latest revised version of the manuscript. Please see Lines 112-113. This is also noted in the Methods (Lines 631-632). We think this will overcome reader’s confusion about the significance level of our statistical results.

8. Lines 310-12: The suggestion that the unfavourable atmospheric conditions in the MDR are related to a strengthening of the Walker Circulation is questionable. The longitude-height cross-section in Figure 5d is for the average over 5S-5N, which is not over the MDR. How are these two related?

-- We thank the reviewer for this comment. In the latest revised version, we added an additional plot (Fig5d) to show the trends in low-level and upper-level winds over the Pacific Ocean basin. The strengthening of the Walker Circulation around the equator (5S-5N) (Fig5e) is clearly embedded in a larger-scale enhancement of zonal overturning circulation in the tropical Pacific. We hope this evidence will help to address the linkage between changes in MDR conditions and the Walker Circulation. We also changed the text (Lines 319-322) to reflect this comment.

REVIEWERS' COMMENTS

Reviewer #2 (Remarks to the Author):

The authors have addressed all my comments and I am happy to recommend acceptance.

Responses for Nature Communications: Paper # NCOMMS-20-46326B

Title: Poleward migration of western North Pacific tropical cyclones related to changes in cyclone seasonality

Reviewers had no comments in the last reviewing process.

Best regards,

Xiangbo Feng, Nicholas Klingaman and Kevin Hodges

REVIEWERS' COMMENTS

Reviewer #2 (Remarks to the Author):

The authors have addressed all my comments and I am happy to recommend acceptance.